# RBPMS inhibits bladder cancer metastasis by downregulating MYC pathway through alternative splicing of ANKRD10

Jingtian Yu[1,2,3,8], Liang Chen [1,2,4,8], Gang Wang [5], Kaiyu Qian [5], Hong Weng[1,6], Zhonghua Yang[1,7], Hang Zheng [1,3,7] ✉ & Mengxin Lu [1,2] ✉

RNA-binding proteins (RBPs) are pivotal mediators of the alternative splicing (AS) machinery of pre-mRNA. Research has demonstrated that the AS process is significantly dysregulated and plays a crucial role in bladder cancer (BLCA). We conducted comprehensive screening and analysis of the TCGA-BLCA cohort, specifically focusing on genes with significant differences in expression levels between carcinoma and adjacent non-cancerous tissues. Among the 500 differentially expressed genes, 5 RNA-binding proteins were identified. Only the RNA-binding protein with multiple splicing (*RBPMS*) demonstrated a consistent downregulation in BLCA and was correlated with an unfavorable prognosis for affected patients. Subsequent experiments revealed that RBPMS exerted inhibitory effects on the epithelial-mesenchymal transition (EMT) pathway and the migratory potential of BLCA cells. RNA-Seq analysis identified *ANKRD10* as a key target mRNA regulated by RBPMS in BLCA. RBPMS depletion in BLCA cells resulted in AS of *ANKRD10* and increased ANKRD10-2 expression. ANKRD10-2 functioned as a transcriptional co-activator of MYC proteins, thereby augmenting their transcriptional activity. Furthermore, *ANKRD10*-2 knockdown significantly rescued the migration enhancement induced by RBPMS depletion in BLCA cells. Taken together, this study revealed a mechanism whereby RBPMS suppresses the migration and invasion of BLCA cells by attenuating MYC pathway activity via the AS of *ANKRD10*.

Alternative splicing (AS) of pre-mRNA occurs in more than 95% of genes, enabling the generation of diverse protein structures and functions from a single gene[1]. The RNA spliceosome is responsible for this process and RNA-binding proteins (RBPs) play a crucial role in binding to specific mRNA[2]. AS has profound effects on cell differentiation and organ development under physiological conditions and is also implicated in various diseases, including cancer, autoimmune disease, and neurodegenerative diseases[3]. Dysregulation of RBPs expression can lead to alterations in target gene splicing, a key mechanism in tumorigenesis and development[4,5]. For instance, the functional role of RNA Binding Fox-1 Homolog 2 (RBFOX2) in AS has been

shown to suppress the metastatic progression of pancreatic cancer[6]. Research indicates that AS is significantly dysregulated in bladder cancer (BLCA) and plays a crucial role in the development, progression, aggressiveness, and therapeutic resistance of this disease[7].

BLCA is the tenth most prevalent malignancy worldwide, exhibiting a male predilection with a fourfold higher incidence rate than that of females. Among males, BLCA is the sixth most common and ninth-most fatal malignancy[8]. Approximately 75% of BLCA cases are diagnosed as non-muscle invasive BLCA (NMIBC). Despite advancements in both surgical and non-surgical interventions, the recurrence rate of NMIBC remains high

[1]Department of Urology, Zhongnan Hospital of Wuhan University, Wuhan, China. [2]Hubei Key Laboratory of Urological Diseases, Wuhan, China. [3]Hubei Clinical Research Center for Laparoscopic/Endoscopic Urologic Surgery, Wuhan, China. [4]Institute of Urology, Wuhan University, Wuhan, China. [5]Department of Biological Repositories, Human Genetic Resources Preservation Center of Hubei Province, Hubei Key Laboratory of Urological Diseases, Zhongnan Hospital of Wuhan University, Wuhan, China. [6]Center for Evidence-Based and Translational Medicine, Zhongnan Hospital of Wuhan University, Wuhan, China. [7]Wuhan Clinical Research Center for Urogenital Tumors, Wuhan, China. [8]These authors contributed equally: Jingtian Yu, Liang Chen. ✉e-mail: zh-urology@whu.edu.cn; mxlu_urology@whu.edu.cn

(50%-90%). Patients with muscle-invasive bladder cancer (MIBC) also have poor prognosis[9]. Additionally, treatment of BLCA poses substantial financial strain, rendering it one of the most expensive cancers to manage[10]. Consequently, unraveling the underlying mechanisms driving BLCA development and identifying targeted therapeutic strategies is crucial for improving the diagnosis and management of this disease. Through analysis of the TCGA-BLCA cohort, we identified that the RNA-binding protein with multiple splicing (*RBPMS*) gene, exhibited low expression in BLCA and was correlated with a poor prognosis. Within the human genome, *RBPMS* gene undergoes AS, resulting in multiple transcript variants that encode at least three distinct protein isoforms. These isoforms are denoted as RBPMS-A, RBPMS-B, and RBPMS-C, respectively[11]. Previous studies have identified the anti-tumorigenic mechanisms of RBPMS in diverse cancer types. RBPMS impedes the proliferation and migratory capabilities of lung cancer cells by facilitating the degradation of *Notch2* mRNA[12]. Notably, RBPMS deficiency reduces the sensitivity of chemotherapeutic drugs in multiple myeloma and ovarian cancer[13–15]. Moreover, RBPMS binds to the AP-1 transcription factor complex, thereby inhibiting its transcriptional activity, which, in turn, suppresses breast cancer proliferation and migration[16].

Ankyrin Repeat Domain 10 (ANKRD10) is a protein belonging to the ankyrin repeat family. Ankyrin repeat proteins typically possess diverse functions in cellular processes, such as protein-protein interactions, transcriptional regulation, and signal transduction[17–19]. For instance, Ankyrin Repeat Domain 1 (ANKRD1), with its dual functionality, is found in the cytoplasm as a constituent of the sarcomere I-band and concurrently functions as a transcriptional cofactor in the nucleus[20]. Moreover, studies have suggested that the leukemogenic potential of Notch ankyrin repeat domains can vary, and this disparity is associated with the activation of Myc[18]. MYC oncoproteins are transcription factors that regulate the transcription of numerous genes in cells, thereby playing a crucial role in biological processes, such as proliferation, differentiation, metastasis, and metabolism. Its significant role as a potent driver of various human cancers has been well documented[21–23].

In this study, we screened the RNA-binding protein RBPMS, which exhibits reduced expression levels in BLCA and is associated with a poor prognosis. RBPMS inhibits BLCA metastasis in vivo and in vitro. Mechanistically, ANKRD10-2 binds to MYC as a transcriptional cofactor and promotes the expression of MYC target genes. RBPMS inhibits malignant tumor progression by binding to and cleaving ANKRD10 mRNA, which reduces translation to ANKRD10-2 and leads to a decrease in MYC transcriptional activity.

## Results

### *RBPMS* expression is diminished in BLCA and is associated with tumor muscle infiltration

Through bioinformatics analysis of the TCGA-BLCA cohort, we analyzed the genomic disparities between BLCA and adjacent tissues. Our findings revealed 250 genes that were highly upregulated in the cancerous milieu and juxtaposed with an equivalent selection of 250 genes that were highly upregulated in the adjacent non-cancerous tissue microenvironment (Supplementary Fig. 1a and Supplementary Table 6). Following comprehensive functional inquiries into these 500 genes in the Gene Ontology (GO) database, we identified five differentially expressed RNA-binding proteins (Supplementary Fig. 1b). Subsequently, we divided the median gene expression into high- and low-expression cohorts and performed Kaplan-Meier survival curve analysis for the aforementioned five genes. Our results showed that the expression levels of *PABPC1L*, *ZEP36*, and *RBPMS* were significantly associated with patient survival prognosis (Supplementary Fig. 1c). Based on the analysis of DEGs in BLCA and patient survival prognosis, we selected *RBPMS* as the study gene. The *RBPMS* expression patterns demonstrated greater consistency with the survival analysis outcomes. *RBPMS* showed increased expression in adjacent cancerous tissues and patients with elevated *RBPMS* expression demonstrated improved prognostic outcomes. Further examination of the TCGA-BLCA cohort data revealed that the mRNA expression levels of *RBPMS* in BLCA were lower

than those observed in normal adjacent tissues across all stages (Supplementary Fig. 1d). Analysis of the GSE13507 cohort within the GEO database corroborated these findings, demonstrating that the mRNA expression level of *RBPMS* in BLCA was consistently lower than that observed in normal adjacent tissues (Fig. 1a). Analysis of the Human Protein Atlas website, specifically examining the results of RBPMS immunohistochemical (IHC) staining in BLCA tissues juxtaposed with normal bladder tissues, revealed a noteworthy contrast. The staining intensity of RBPMS in BLCA tissues appeared faint, whereas that in the epithelium of normal bladder tissue was markedly intense. These observations strongly suggest that the protein level of RBPMS was notably higher in the epithelium of normal bladder tissue (Supplementary Fig. 1e).

*RBPMS* mRNA expression was consistently lower in various cancers than in the adjacent normal tissues (Supplementary Fig. 2a). Analysis of BLCA-related cohorts in the GEO database revealed a distinct pattern in RBPMS expression levels. Specifically, RBPMS expression was higher in NMIBC cases, whereas it was lower in MIBC cases (Fig. 1b, c and Supplementary Fig. 2b–e). Furthermore, as tumor staging advanced, a consistent downward trend in RBPMS expression was evident, suggesting a gradual decrease in its expression level with the progression of BLCA staging (Fig. 1d). IHC staining was performed on both paracancerous tissue and BLCA tissue. Our findings revealed a marked decrease in RBPMS expression in BLCA tissue, with further reduction as the tumor infiltrated the muscle layer (Fig. 1e). We categorized all tissues into cohorts exhibiting either high or low RBPMS expression levels. RBPMS was highly expressed in 81.8% of the paracancerous tissues and 46.0% of the cancerous tissues. The rate of high expression in paracancerous tissues was significantly higher than that in bladder cancer tissues (OR: 5.276, 95% CI: 1.054–26.402, $p = 0.028$). Moreover, the high expression rate of RBPMS was significantly higher in NMIBC tissues than MIBC tissues (77.8% vs 33.3%, OR: 7.000, 95% CI: 1.940–25.255, $p = 0.002$). Conversely, no statistically significant disparities were discerned in RBPMS expression levels across varying pathological grades, lymph node metastasis statuses, or tumor dimensions (Table 1). We conducted a comprehensive functional enrichment analysis of the proteins associated with RBPMS using the STRING database. Our findings revealed that RBPMS-related proteins play a significant role in key pathways such as RNA splicing and mRNA progression (Fig. 1f). According to the NCBI database, *RBPMS* exhibits four splicing variants, two of which have identical coding sequence (CDS) regions. Consequently, RBPMS showed three distinct protein isoforms (Fig. 1g). Analysis of the TCGA-BLCA cohort revealed that the bladder primarily expressed the RBPMS-1 and RBPMS-3 splicing variants (Fig. 1h). This observation was corroborated by the results of quantitative reverse-transcription PCR (qRT-PCR) analysis of BLCA cell lines. Notably, among the examined cell lines, T24 and UM-UC-3 showed the lowest expression levels of RBPMS (Fig. 1i). These findings suggested a downregulation of RBPMS expression in BLCA, which exhibited a negative correlation with the malignancy of this disease.

### RBPMS inhibits the metastasis of BLCA cells both in vitro and in vivo

*RBPMS-1/4* and *RBPMS-3* encode proteins designated RBPMS-A and RBPMS-C, respectively. We generated T24 cell lines stably overexpressing RBPMS-A or RBPMS-C. Subsequent transwell experiments demonstrated a significant reduction in the number of T24 cells passing through the compartments after overexpression of RBPMS-A or RBPMS-C (Fig. 2a). Additionally, the results of the wound healing experiment revealed a marked decrease in the migration rate of T24 cells after overexpression of RBPMS-A or RBPMS-C (Fig. 2b). Gene Set Enrichment Analysis (GSEA) of the GSE32548 and GSE32894 cohorts from the GEO database revealed significant enrichment of genes associated with the epithelial-mesenchymal transition (EMT) pathway in samples with low expression levels of RBPMS (Fig. 2c and Supplementary Fig. 3a). Overexpression of either RBPMS-A or RBPMS-C in T24 cells resulted in significant downregulation of N-cadherin and Vimentin protein levels. Conversely, upregulation of E-cadherin protein levels was observed (Fig. 2d). Given the comparatively elevated

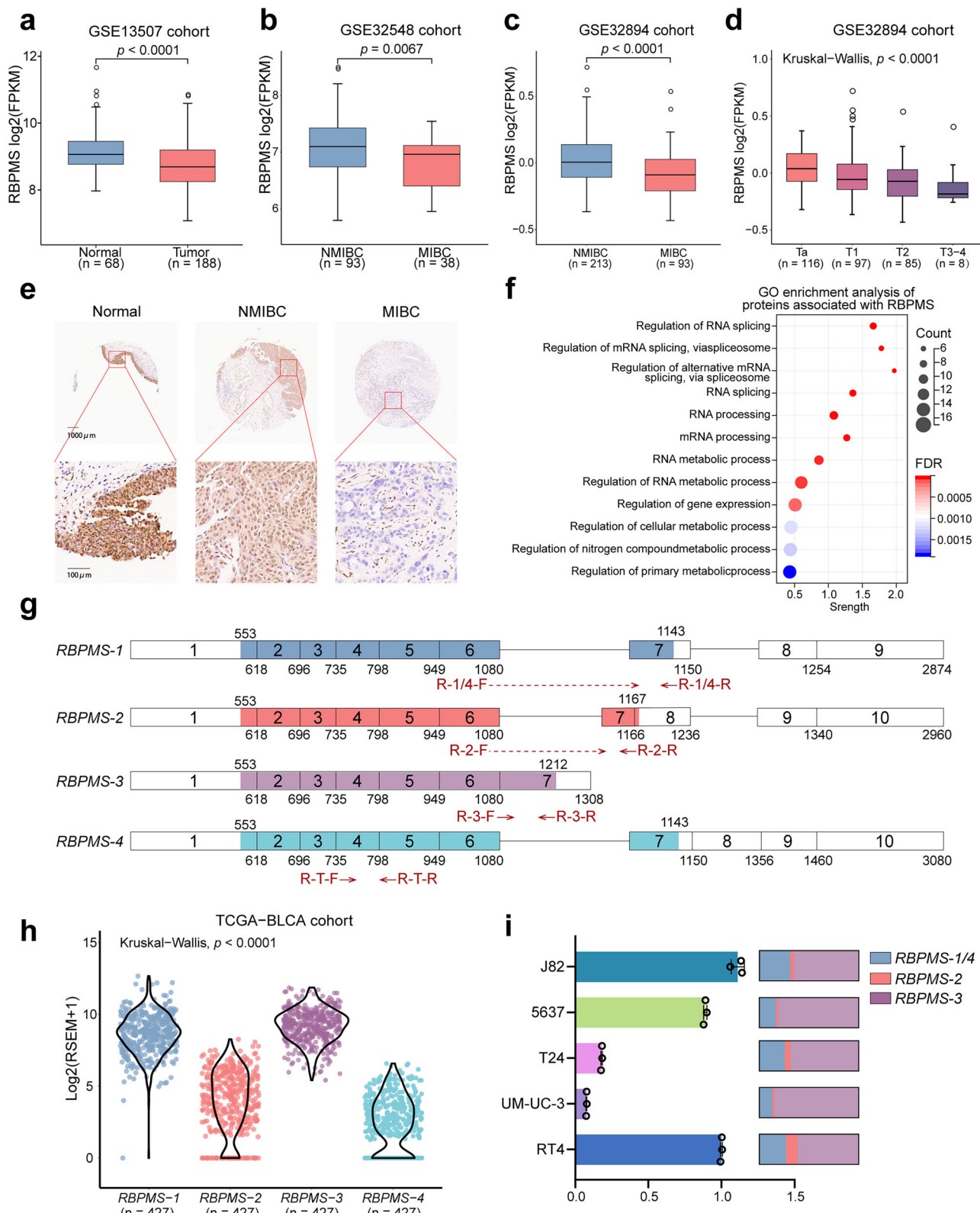

**Fig. 1 | RBPMS is expressed at lower levels in MIBC than in NMIBC. a** mRNA levels of *RBPMS* in BLCA (*n* = 188) and normal tissues (*n* = 68) in the GSE13507 cohort. **b** mRNA levels of *RBPMS* in MIBC (*n* = 38) and NMIBC (*n* = 93) in the GSE32548 cohort. **c** mRNA levels of *RBPMS* in MIBC (*n* = 93) and NMIBC (*n* = 213) in the GSE32894 cohorts. **d** mRNA levels of *RBPMS* in various T-stages of BLCA in the GSE32894 cohort. **e** Representative images of IHC staining of RBPMS in normal tissue, NMIBC and MIBC from tissue microarrays. **f** Functional enrichment analysis of proteins interacting with RBPMS (https://cn.string-db.org/). **g** Schematic representation of the four splice variants of *RBPMS* mRNA. Rectangles represent exons and colors represent coding sequence regions. "R-1/4," "R-2," and "R-3" arrows indicate the locations of RT-PCR primers for determining different isoforms expression. "R-T" arrows indicate the positions of qRT-PCR primers used to determine total *RBPMS* expression. **h** mRNA expression levels of four splice variants of RBPMS in the TCGA-BLCA cohort (*n* = 427). **i** mRNA expression levels of the four splice variants of RBPMS in BLCA cell lines were detected by qRT-PCR (*n* = 3). Data are shown as mean ± SD. The *n* number represents n biologically independent experiments in each group.

**Table 1 | Clinicopathological characterisation of RBPMS expression in the HBlaU079Su01 cohort**

|  | Variables | RBPMS expression, n (%) | | p value | Statistics method |
|---|---|---|---|---|---|
|  |  | High (>80%) | Low (≤80%) |  |  |
| Cancer and paracancer | Paracancer | 9 (81.82) | 2 (18.18) | 0.0284 | Chi-square |
|  | Cancer | 29 (46.03) | 34 (53.97) |  |  |
| Muscle invasion | NMIBC | 14 (77.78) | 4 (22.22) | 0.0016 | Chi-square |
|  | MIBC | 14 (33.33) | 28 (66.67) |  |  |
| Pathological grading | Low grade | 4 (57.14) | 3 (42.86) | 0.9252 | Chi-square with Yates' correction |
|  | High grade | 25 (47.17) | 28 (52.83) |  |  |
| N stage | N0 | 22 (48.89) | 23 (51.11) | 0.3868 | Chi-square with Yates' correction |
|  | N ≥ 1 | 2 (25.00) | 6 (75.00) |  |  |
| Tumor size | ≤ 3 cm | 11 (61.1) | 7 (38.9) | 0.2197 | Chi-square |
|  | > 3 cm | 17 (43.6) | 22 (56.4) |  |  |

*Tumor size* The longest diameter, cm, *NMIBC* Non-muscular invasive bladder cancer, *MIBC* Muscular invasive bladder cancer.

expression levels of RBPMS in the 5637 cell line within BLCA, we selected this specific cell line to construct *RBPMS* knockout cell lines (Supplementary Fig. 3b). Following *RBPMS* knockout, an increase in N-cadherin and Vimentin protein levels was observed concomitant with a decline in E-cadherin levels. Subsequent overexpression of either RBPMS-A or RBPMS-C in *RBPMS* knockout cell lines resulted in a modest recovery in the levels of EMT-related proteins (Supplementary Fig. 3c). Additional experiments using transwell and wound healing techniques further corroborated these findings, indicating that *RBPMS* knockout resulted in an increased migration rate of 5637 cells. In contrast, overexpression of either RBPMS-A or RBPMS-C significantly reduced the migration rate of these cells (Supplementary Fig. 3d-e).

Subsequently, we constructed a lung metastasis model in nude mice by tail vein injection of cancer cells. We assessed the effect of RBPMS-A or RBPMS-C overexpression on the migration ability of BLCA cells in vivo using this experimental animal metastasis model[24–27]. Six weeks post-injection of BLCA cells into the tail vein of nude mice, lung metastasis was visualized using in vivo fluorescence imaging. The results indicated a significant reduction in the number of cancer cells that spread to the lungs following the overexpression of RBPMS-A or RBPMS-C (Fig. 2e, f). After euthanizing the nude mice, lung tissues were analyzed using H&E staining and IHC analysis. The results showed that overexpression of RBPMS-A or RBPMS-C significantly reduced the infiltration range of cancer cells within the lung tissue (Fig. 2g). These findings provided compelling evidence demonstrating the inhibitory role of RBPMS in BLCA metastasis both in vivo and in vitro.

## RBPMS induces AS of *ANKRD10* mRNA in BLCA

Considering that RBPMS has two transcripts and exhibits functional similarity, to comprehensively understand the molecular mechanism underlying RBPMS-mediated inhibition of BLCA metastasis, we conducted RNA-Sequencing (RNA-seq) analysis on *RBPMS* knockout and control cell lines. To avoid off-target effects of sgRNA and potential variability in RNA-seq experiments, we conducted differential gene expression and intersection analyses of the RNA-Seq data obtained from comparisons between sgRBPMS-1 and sgCtrl, as well as sgRBPMS-2 and sgCtrl, respectively (Fig. 3a, b and Supplementary Fig. 4a). Subsequently, GO and KEGG pathway enrichment analyses were performed on the common differentially expressed genes (DEGs) (Fig. 3c and Supplementary Fig. 4b). GO analysis of the identified DEGs revealed the association of RBPMS with diverse cellular processes including negative regulation of cell migration, protein binding, and transcriptional regulation. This finding was consistent with our previous observations of the inhibitory role of RBPMS in BLCA migration and invasion. Subsequent analysis of intracellular mRNA splicing alterations due to *RBPMS* knockout revealed a pronounced occurrence of skipped exon

(SE) splicing events, as evidenced by the intersection of the sgRBPMS-1 and sgRBPMS-2 results (Fig. 3d and Supplementary Fig. 4c). Reverse transcription-PCR (RT-PCR) analysis of these genes, subsequent to *RBPMS* knockout in 5637 cells, revealed alterations in seven genes. Furtherly, overexpression of RBPMS-A or RBPMS-C could rescue this regulatory effect on ANKRD10 (Fig. 3e). Quantitative analysis of RT-PCR bands revealed density variations among splicing variants, demonstrating that RBPMS exhibited the most pronounced regulatory effect on *ANKRD10* transcript variant expression (Fig. 3f).

ANKRD10 is characterized by two splicing variants: a relatively elongated ANKRD10-1 variant and a comparably shorter ANKRD10-2 variant (Fig. 3g). On the RBPsuite website, we predicted possible binding sites for RBPMS on *ANKRD10-1* and *ANKRD10-2* mRNA[28]. ANKRD10-1 and ANKRD10-2 both have the highest scores at the 420–490 base pairs (bp) locus (Fig. 3h and Supplementary Fig. 4d). The RNA immunoprecipitation (RIP) experiment using RBPMS antibodies yielded quantifiable results via qRT-PCR, confirming the interaction between RBPMS and *ANKRD10* mRNA (Fig. 3i). Western blot analyses demonstrated an increase in ANKRD10-2 levels and a significant decrease in ANKRD10-1 levels following *RBPMS* knockout. Conversely, the overexpression of RBPMS-A or RBPMS-C resulted in the opposite effect (Fig. 3j). Through analysis of predicted RBPMS binding sites on *ANKRD10-1* RNA, we designed primers targeting high-scoring sequence fragments and performed crosslinking immunoprecipitation (CLIP) experiments to evaluate direct interactions between RBPMS protein and *ANKRD10-1*. The experimental findings aligned with both computational predictions data, demonstrating that RBPMS specifically interacts with two distinct regions of *ANKRD10-1* RNA: the 420–490 bp locus and 1540–1610 bp locus (Fig. 3k). In the TCGA-BLCA cohort, the expression of ANKRD10-2 was found to be significantly higher in BLCA than in normal bladder tissue. This was in contrast to the observed expression pattern of RBPMS in the same cohort (Supplementary Fig. 4e). Although ANKRD10-1 expression in BLCA was slightly reduced compared to that in normal bladder tissue, the difference was not statistically significant (Supplementary Fig. 4f). These findings suggest that in BLCA tissues with low RBPMS expression, ANKRD10-2 constitutes a larger proportion of the total ANKRD10 (Fig. 3l). Conversely, the expression of ANKRD10-1 correlated positively with RBPMS (Supplementary Fig. 4g). Co-immunoprecipitation (Co-IP) experiments revealed the interaction of ESPR1, a type-specific splicing regulatory factor for epithelial cells, with RBPMS (Supplementary Fig. 4h). Subsequently, we assessed the transcriptional levels of *ANKRD10*-1 and *ANKRD10*-2, as well as the protein levels of RBPMS in various BLCA cell lines (Supplementary Fig. 4i). The results revealed a positive correlation between *ANKRD10-1* mRNA and RBPMS protein expression, whereas *ANKRD10-2* mRNA was negatively correlated with RBPMS protein expression (Supplementary Fig. 4j, k). These

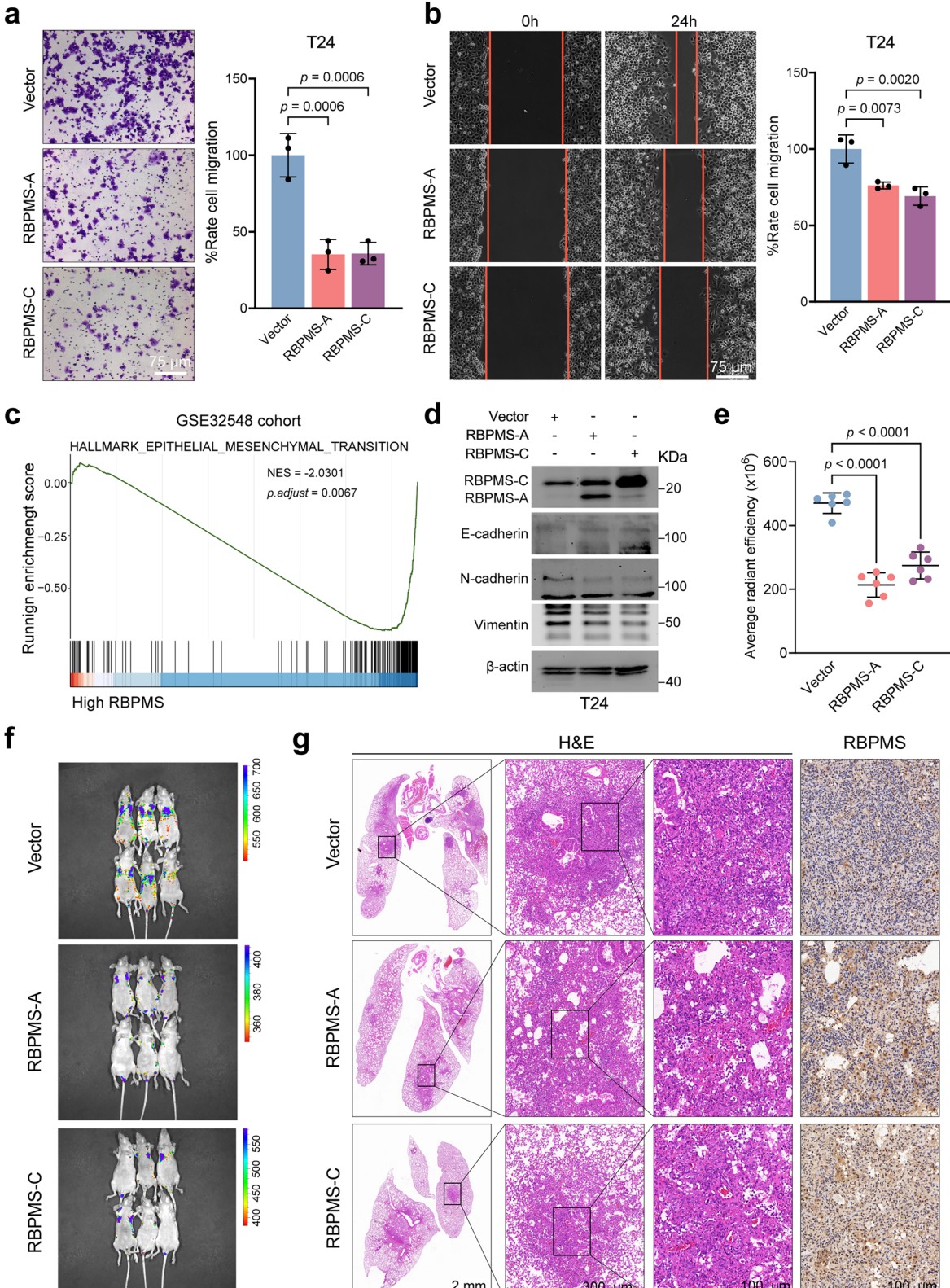

**Fig. 2 | RBPMS inhibits the metastasis of BLCA cells in vitro and in vivo.**
**a** Representative image (left) of transwell assays and cell counting histogram (right) after overexpression of RBPMS-A or RBPMS-C in T24 cells ($n = 3$). **b** Representative image (left) of wound healing assays and cell counting histogram (right) after overexpression of RBPMS-A or RBPMS-C in T24 cells ($n = 3$). **c** GSEA of the GSE32548 cohort shows that RBPMS is negatively related to epithelial mesenchymal transition pathway. **d** Immunoblot assays showing the expression of EMT pathway-related proteins after RBPMS-A or RBPMS-C overexpression in T24 cells. The average radiation efficiency (**e**) and fluorescence imaging (**f**) in the lungs of nude mice 6 weeks after tail vein injection of T24 cells stably expressing RBPMS-A or RBPMS-C ($n = 6$). **g** Representative images of H&E- and IHC-stained lung sections from lung metastasis model mice. Data are shown as mean ± SD. The $n$ number represents n biologically independent experiments in each group.

**Fig. 3 | RNA-seq reveals *ANKRD10* as a splicing target of RBPMS. a** RNA-seq analysis after knockout of *RBPMS* in 5637 cells and differential gene analysis by R package "Desep2." **b** Heatmap illustrating the gene expression profiles of the 5637 sgCtrl and sgRBPMS-1 cell lines. **c** GO enrichment analysis of DEGs in RNA-seq results after knockout of *RBPMS*. **d** Volcano plot showing the inclusion (Inc) level differences and *p*-values for differential AS events between sgCtrl and sgRBPMS 5637 cell lines. SE, skipped exon; MXE, mutually exclusive exon; RI, retained intron; A3SS, alternative 3' splice site; A5SS, alternative 5' splice site. **e** Validation of AS events of representative genes in RNA-seq results by RT-PCR after knockdown or overexpression of RBPMS in 5637 cells. **f** Quantitative densitometric analysis of RT-PCR band intensities for AS events is presented in the heat map, with β-actin serving as the normalization control. **g** Schematic representation of the *ANKRD10* splice variants. "Splicing" arrows indicate the locations of RT-PCR primers for determining different isoforms expression. "RIP" arrow indicates the position of the qRT-PCR primer used to determine *ANKRD10* expression. **h** Predicted RBPMS binding site scores on *ANKRD-1* mRNA on the RBPsuite website (http://www.csbio.sjtu.edu.cn/bioinf/RBPsuite/). The horizontal coordinate each segment represents a nucleotide fragment of 70 bp size. **i** RIP assay was performed in 293 T cell lysates using anti-RBPMS or anti-IgG, and enrichment of *ANKRD10* was detected by qRT-PCR (*n* = 3). **j** Immunoblot showing the expression of ANKRD10-1 and ANKRD10-2 protein after *RBPMS* knockout or overexpression in 5637 cells. **k** CLIP experiments were performed to detect direct binding between RBPMS protein and *ANKRD10-1*, with sequence fragments selected for validation based on high prediction scores from figure (**h**), while sequence fragments with low scores were used as negative controls (*n* = 3). **l** High (*n* = 214) and low (*n* = 213) grouping by median *RBPMS* expression in the TCGA-BLCA cohort, showing the proportion of ANKRD10-2 to total ANKRD10. Data are shown as mean ± SD. The *n* number represents n biologically independent experiments in each group.

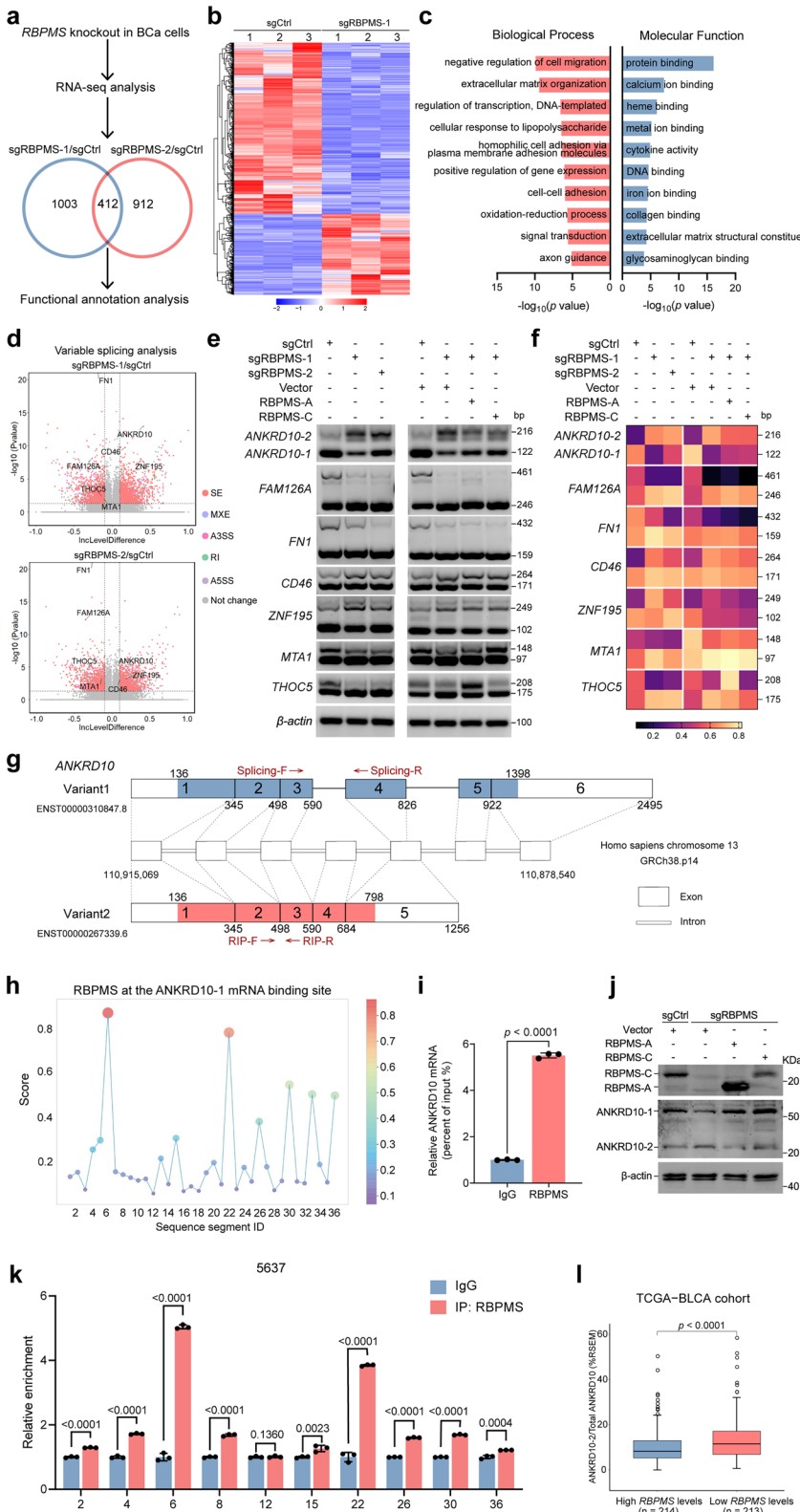

findings collectively suggested that RBPMS exerted an inhibitory effect on BLCA metastasis through the cleavage of *ANKRD10*.

**RBPMS exerts inhibitory effects on MYC target gene expression**
We studied the precise molecular mechanisms underlying the effects of RBPMS and ANKRD10 on BLCA by manipulating the ANKRD10-1 and ANKRD10-2 levels in 5637 cells. We knocked down *ANKRD10-1* or overexpressed ANKRD10-2 in 5637 cells and knocked down *ANKRD10-2* or overexpressed ANKRD10-1 in 5637 cells previously knocked out of *RBPMS*. Subsequent RNA-Seq analyses were performed, followed by GSEA within the Hallmark gene cluster. After GSEA enrichment analysis, the ten most significant gene sets were identified. Subsequently, we intersected these gene sets with the RNA-Seq results obtained from *RBPMS* knockout experiments (Fig. 4a).

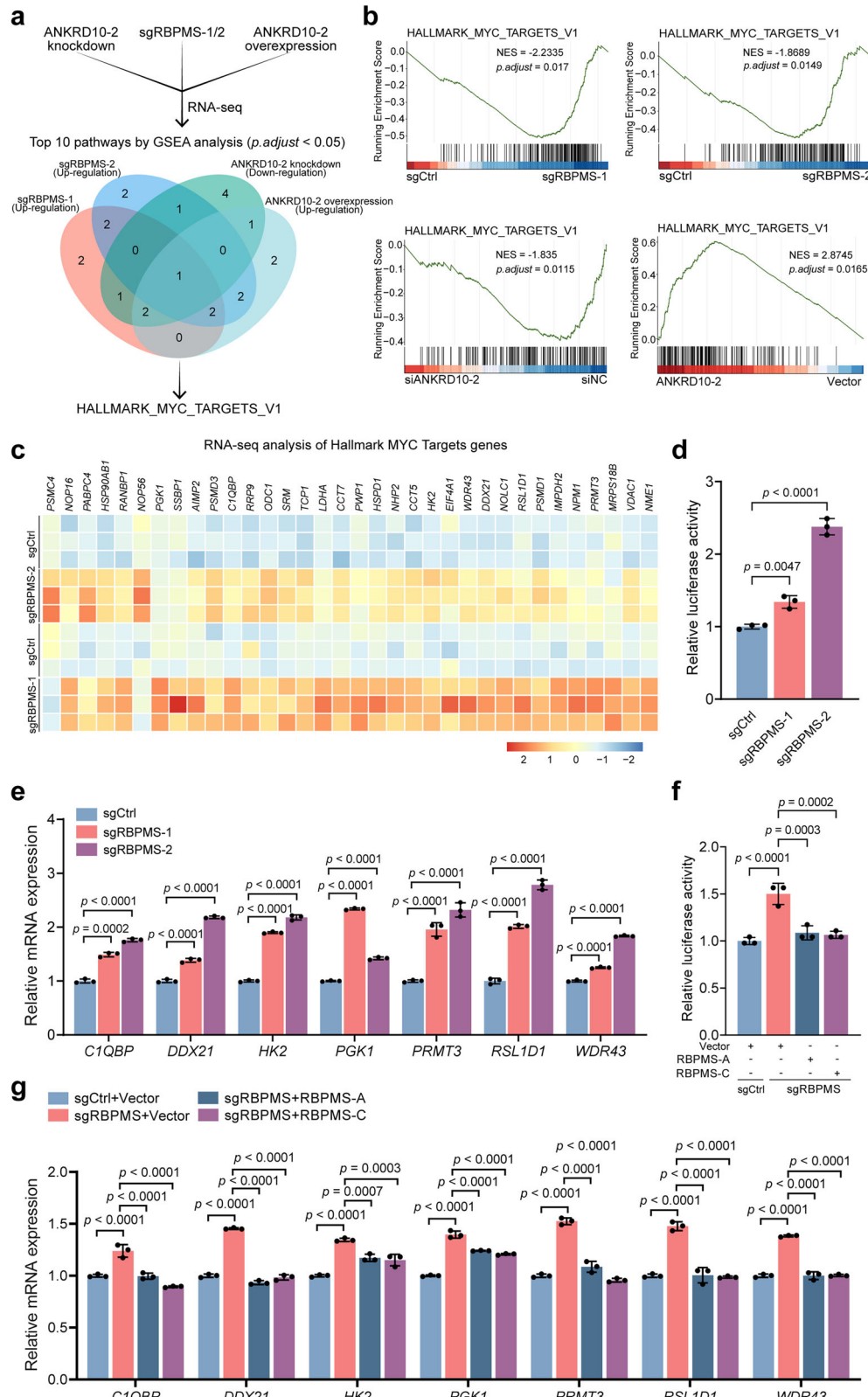

**Fig. 4 | RBPMS and ANKRD10 affect BLCA progression by regulating MYC target gene expression. a** RNA-seq data were analyzed using GSEA enrichment analysis, and pathways upregulated by *RBPMS* knockout and *ANKRD10*-2 overexpression were intersected with pathways downregulated by *ANKRD10*-2 knockdown. The pathways obtained by GSEA enrichment were selected as 10 with the smallest p.adjust values. **b** Hallmark_MYC_Targets_V1 gene set demonstrated by GSEA enrichment of all RNA-seq results. **c** Heatmap of Hallmark_MYC_Targets_V1 genes in RNA-seq results of knockout *RBPMS*. **d** 5637 sgCtrl and sgRBPMS cells were transfected with the E-box dual-luciferase plasmid and then assayed for reporter fluorescence activity (*n* = 3). **e** Validation of differential mRNA levels of MYC target genes in RNA-seq assays after *RBPMS* knockout by qRT-PCR (*n* = 3). **f** After overexpression of RBPMS-A or RBPMS-C, 5637 sgCtrl and sgRBPMS cells were transfected with the E-box dual-luciferase plasmid and assayed for reporter fluorescence activity (*n* = 3). **g** mRNA levels of MYC target genes in 5637 sgCtrl and sgRBPMS cells detected by qRT-PCR after overexpression of RBPMS-A or RBPMS-C (*n* = 3). Data are shown as mean ± SD. The *n* number represents n biologically independent experiments in each group.

GSEA results revealed that the gene set enriched by both RBPMS and ANKRD10-2 corresponded to MYC targets. Notably, *RBPMS* knockout or overexpression of ANKRD10-2 resulted in significant upregulation of MYC target gene expression (Fig. 4b, c and Supplementary Fig. 5a–d). In contrast, RBPMS and ANKRD10-1 did not intersect this pathway (Supplementary Fig. 5e). MYC can form heterodimeric complexes with transcription factor MAX. These complexes bind to specific DNA sequences known as enhancer boxes (E-boxes), thereby orchestrating transcriptional regulation of downstream target genes[29,30]. We used the E-box dual-luciferase plasmid in our study, as used in previous studies, which incorporated duplicate E-box sequences within the luciferase reporter construct[31]. This enabled us to evaluate the transcriptional effect of MYC on target genes using relative fluorescence intensity measurements. Following transfection of the E-box dual luciferase plasmid into both 5637 and control cell lines following *RBPMS* knockout, a significant increase in fluorescence intensity was observed (Fig. 4d). Subsequent qRT-PCR analysis revealed a significant increase in the transcription level of MYC target genes after *RBPMS* knockout (Fig. 4e). Moreover, upon knockout of *RBPMS*, we overexpressed either RBPMS-A or RBPMS-C, which resulted in a decline in the transcriptional activity of MYC and a significant reduction in the transcriptional levels of downstream target genes (Fig. 4f, g). GSEA enrichment analysis of BLCA chips from TCGA and GEO revealed that MYC target genes were enriched in the low RBPMS expression group (Supplementary Fig. 5f). Our findings were supported by correlation analysis between RBPMS and MYC target genes in both BLCA and normal bladder tissues from TCGA and GTEx datasets. Specifically, a negative correlation was observed between the expression levels of RBPMS and MYC target genes (Supplementary Fig. 5g). These results strongly suggested an inverse relationship between RBPMS and MYC target genes, highlighting the influence of RBPMS expression on the transcriptional regulation of downstream target genes by MYC.

### The interaction between ANKRD10-2 and MYC facilitates transcriptional activation of its target genes

Given the observed promotion of *ANRKD10* precursor mRNA cleavage into *ANKRD10-2* upon *RBPMS* knockout coupled with the enhanced transcriptional activity of MYC, our subsequent investigation aimed to elucidate the effect of ANKRD10-2 on MYC transcription. After *ANKRD10-2* knockdown, GSEA enrichment analysis of the RNA-Seq data within the GO gene set revealed a significant association between ANKRD10-2 and coregulation and activation (Fig. 5a, b). *ANKRD10-2* knockdown in RBPMS-depleted 5637 cells significantly reduced the transcriptional activity of MYC towards target genes, as demonstrated by luciferase reporter assays (Fig. 5c). Furthermore, qRT-PCR analysis revealed a significant downregulation in the transcription levels of MYC-related target genes upon *ANKRD10-2* knockdown (Fig. 5d). After transfection of the ANKRD10-2 plasmid into 5637 cells, luciferase reporter assays and qRT-PCR analyses were conducted. The results revealed that ANKRD10-2 facilitated the transcriptional activity of MYC and consequently increased the mRNA levels of downstream target genes (Fig. 5e, f). Studies have reported that ANKRD1 functions as a transcriptional cofactor in the nucleus[20]. The Ankyrin Repeat Domain is also implicated in MYC activation[18]. Subsequently, exogenous MYC, ANKRD10-1, and ANKRD10-2 plasmids were introduced into 293 T cells and Co-IP experiments were conducted. These findings revealed a specific interaction between MYC and ANKRD10-2, with no discernible interaction between MYC and ANKRD10-1 (Fig. 5g, h). To evaluate the influence of ANKRD10-2 on MYC's capacity to bind target gene promoter regions, we conducted chromatin immunoprecipitation (ChIP) analyses following *ANKRD10-1* and *ANKRD10-2* knockdown in 5637 cells. Our findings demonstrated that ANKRD-2 depletion resulted in a broad reduction of MYC binding to target gene promoter regions, with HK2 target gene exhibiting the most significant decrease, aligning with our earlier quantitative PCR (qPCR) observations. In contrast, ANKRD10-1 depletion

demonstrated minimal effect on MYC's association with target gene promoters (Fig. 5i). Our findings suggested a potential role for ANKRD10-2 as a transcriptional co-activator, wherein it interacted with MYC to enhance the MYC-mediated transcription of target genes.

### RBPMS curtails BLCA metastasis by cleaving *ANKRD10* mRNA, thereby impeding MYC transcription

Next, we determined whether RBPMS affects BLCA metastasis through the cleavage of *ANKRD10* mRNA, thereby impeding the activation of MYC target genes. Previous studies have demonstrated that following *RBPMS* knockout, there is a notable increase in the migratory capacity of BLCA cells. To further elucidate the underlying mechanisms, we conducted knockdown experiments targeting *ANKRD10-2* in *RBPMS* knockout cells. The results from the wound healing experiments revealed a significant reduction in the migration enhancement observed in *RBPMS* knockout BLCA cells when *ANKRD10*-2 was concurrently knocked down (Fig. 6a, b). The findings from the transwell experiment further supported our observations, indicating that *ANKRD10*-2 knockdown significantly reduced the effect of *RBPMS* knockout on the migration ability of cell lines (Fig. 6c, d). The luciferase reporter experiment revealed that *ANKRD10*-2 knockdown influenced the augmented MYC transcriptional activity resulting from the *RBPMS* knockout (Fig. 6e). Furthermore, qRT-PCR results indicated that *ANKRD10*-2 knockdown increased the transcription levels of MYC target genes triggered by *RBPMS* knockout (Fig. 6f). In summary, our findings suggested that RBPMS suppressed *ANKRD10* mRNA splicing to ANKRD10-2, thereby influencing the interaction between ANKRD10-2 and MYC. This interaction ultimately impeded MYC transcriptional regulation of its target genes, thereby exerting an inhibitory effect on BLCA metastasis. This intricate regulatory cascade highlights the pivotal role of RBPMS in modulating the critical molecular pathways involved in the progression of BLCA.

In summary, our research revealed a significant downregulation of RBPMS expression in BLCA, particularly in cases characterized by muscle layer infiltration, which correlated with poorer patient prognosis. Functionally, RBPMS inhibited BLCA cell migration both in vivo and in vitro. Mechanistically, RBPMS functions by binding to the ANKRD10 precursor mRNA, thereby impeding the cleavage process that generates ANKRD10-2. This interruption disrupts the interaction between ANKRD10-2 and MYC, leading to reduced transcriptional activation. Consequently, downstream expression of MYC target genes was suppressed, ultimately contributing to the inhibition of BLCA metastasis (Fig. 6g).

## Discussion

In our study, we screened five RNA-binding protein genes by analyzing 500 genes with the most significant differences in high or low expression between cancer and paracancer tissues in the TCGA-BLCA cohort. Further survival prognosis analysis revealed that only the outcomes associated with RBPMS were significantly different and aligned with the observed differential expression. Further evaluation of BLCA microarrays in the GEO database confirmed reduced expression of RBPMS in BLCA, particularly in MIBC. Previous investigations have indicated that RBPMS-A and RBPMS-C mRNA and protein levels are commonly detected in ovarian cancer, whereas RBPMS-B mRNA and protein expression levels are undetectable[32]. Consistently, our assay conducted in BLCA yielded similar findings. Moreover, in vivo and in vitro experiments demonstrated that both RBPMS-A and RBPMS-C inhibited BLCA cell migration and invasion.

As an RNA binding protein, RBPMS has been reported to play an important role in RNA splicing regulation in many studies[33]. Gan et al. reported that RBPMS involved in smooth muscle cell specific alternative splicing of Tpm1 exon 3 in cell-free assays[34]. In addition, RBPMS was reported as an important regulator of cardiomyocyte contraction and cardiac function through RNA alternative splicing[35]. To explore the effect of RBPMS on regulating mRNA processing within BLCA, we conducted

**Fig. 5 | ANKRD10-2 interacts with MYC to promote its transcriptional regulatory function on target genes. a** GSEA enrichment of RNA-seq data after *ANKRD10* knockdown in 5637 sgRBPMS cells demonstrates a positive correlation with ANKRD10-2 in the transcriptional coactivator and coregulator activity pathways. **b** *ANKRD10-2* knockdown in 5637 sgRBPMS cells was subjected to RNA-seq and enriched by GSEA using the GO database. **c** After *ANKRD10-2* knockdown, 5637 sgRBPMS cells were transfected with an E-box dual-luciferase plasmid and assayed for reporter fluorescence activity (*n* = 3). **d** mRNA levels of MYC target genes in 5637 sgRBPMS cells were detected by qRT-PCR after *ANKRD10-2* knockdown (*n* = 3). **e** After *ANKRD10-2* overexpression, 5637 sgRBPMS cells were transfected with an E-box dual-luciferase plasmid and assayed for reporter fluorescence activity (*n* = 3). **f** mRNA levels of MYC target genes in 5637 cells were detected by qRT-PCR after *ANKRD10-2* overexpression (*n* = 3). Transfection of 293 T cells with the specified plasmids lasted 48 h, followed by Co-IP using either anti-FLAG (**g**) or anti-HA antibodies (**h**). **i** ChIP-qPCR analysis examined MYC binding to promoters of *C1QBP*, *DDX21*, *HK2*, *PGK1*, *PRMT3*, *RSL1D1* and *WDR43* in 5637 cells after *ANKRD10-1* or *ANKRD10-2* knockdown (*n* = 3). Data are shown as mean ± SD. The *n* number represents n biologically independent experiments in each group.

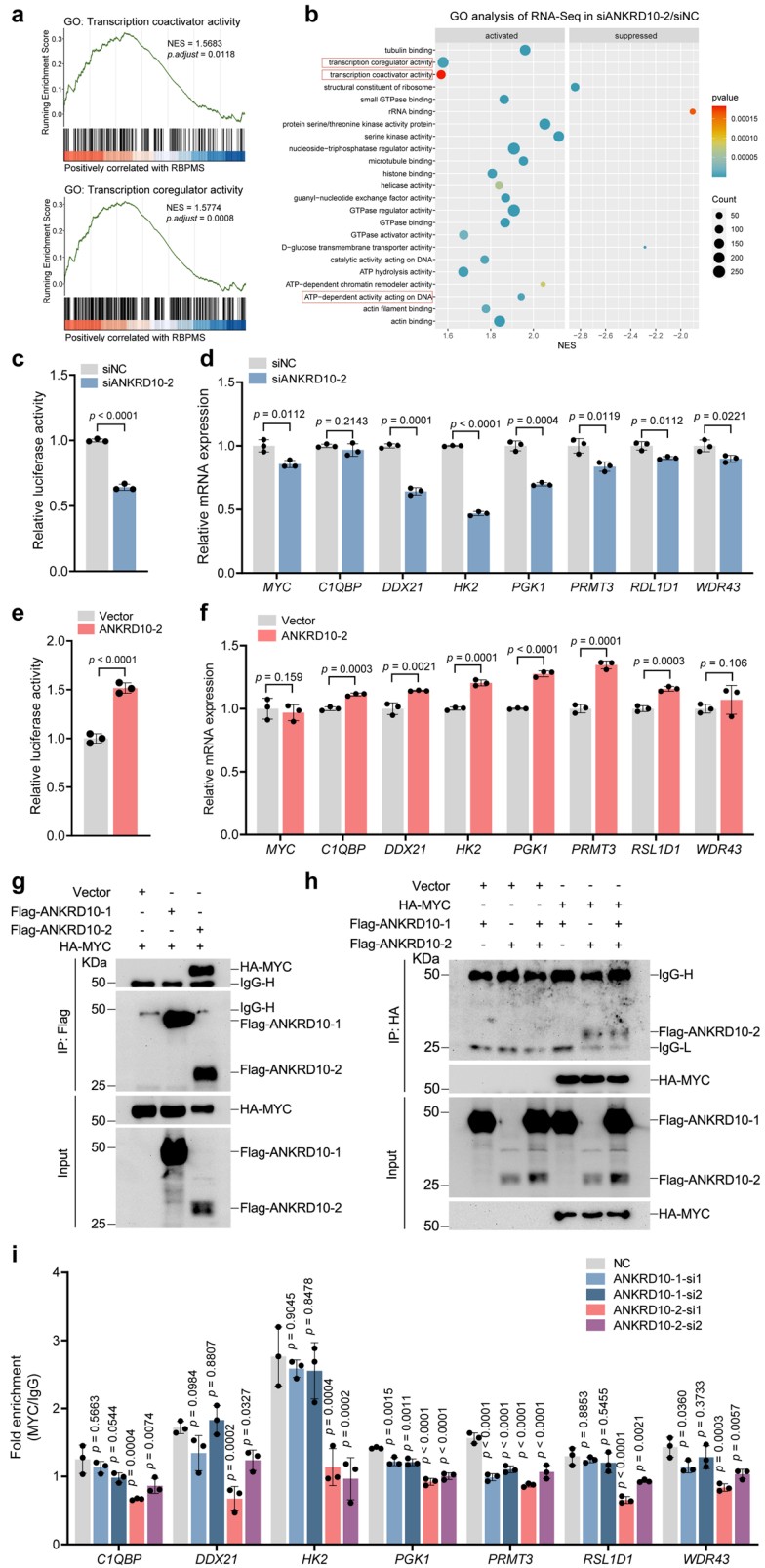

RNA-seq analysis following *RBPMS* knockout in BLCA cells[11]. Through analysis and verification of AS events in the sequencing results, we identified *ANKRD10* as being significantly regulated by RBPMS.

ANKRD10 has two primary transcript variants, ANKRD10-1, which encodes a longer sequence, and ANKRD10-2, which encodes a shorter sequence. Our findings indicate that a deficiency in RBPMS results in

increased mRNA expression levels of *ANKRD10-2*. Moreover, ANKRD10-2 was expressed at higher levels in the BLCA tissues than in the paraneoplastic tissues in the TCGA-BLCA cohort.

Comprehensive analysis of transcriptome sequencing results pertaining to RBPMS and ANKRD10 revealed a potential interplay between RBPMS and ANKRD10-2 in influencing cancer progression, possibly

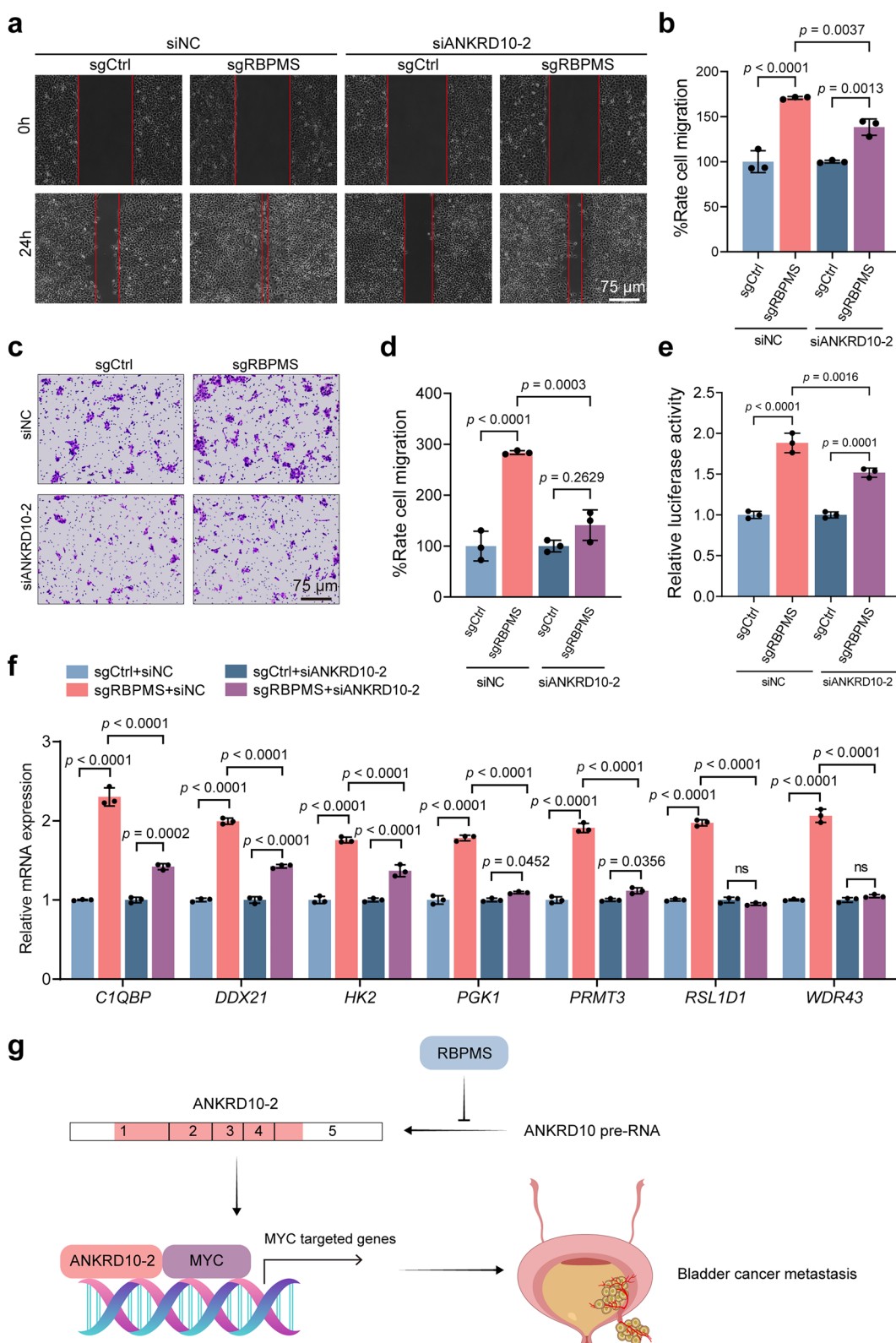

**Fig. 6 | RBPMS inhibits MYC transcription of target genes by AS of ANKRD10.** Representative image (**a**) and cell counting histogram (**b**) of wound healing from the indicated groups with *ANKRD10-2* knockdown in 5637 sgCtrl and sgRBPMS cells (*n* = 3). Representative image (**c**) and cell counting histogram (**d**) of transwell assays assays from the indicated groups with *ANKRD10-2* knockdown in 5637 sgCtrl and sgRBPMS cells (*n* = 3). **e** After *ANKRD10-2* knockdown, 5637 sgCtrl and sgRBPMS cells were transfected with an E-box dual-luciferase plasmid and assayed for reporter fluorescence activity (*n* = 3). **f** mRNA levels of MYC target genes in 5637 sgCtrl and sgRBPMS cells detected by qRT-PCR after *ANKRD10*-2 overexpression (*n* = 3). **g** Schematic showing the mechanism by which RBPMS inhibits BLCA migration and invasion via the RBPMS /ANKRD10/MYC axis. Data are shown as mean ± SD. The *n* number represents n biologically independent experiments in each group.

through their influence on MYC target gene expression. However, studies on the role of ANKRD10 in cancer are limited. Previous studies have demonstrated that Notch ankyrin repeat domains play a pivotal role in the transcriptional activation of MYC[18].

Our experimental findings corroborate that ANKRD10-2, rather than ANKRD10-1, can bind to MYC and enhance its transcriptional activity, consequently leading to upregulation of downstream target genes. AS plays a key role in tumor progression and development[36]. Current research focuses on developing molecular inhibitors targeting oncogenic splicing factors and machinery components as therapeutic interventions[37]. A prime example is androgen receptor splice variant 7 (AR-V7), which maintains transcriptional activity despite lacking the ligand-binding domain targeted by enzalutamide and abiraterone. This characteristic leads to treatment resistance in prostate cancer patients[38,39]. Recent research has identified SC912, a small molecule that binds to both full-length AR and AR-V7 through the AR N-terminal domain, thereby inhibiting AR-V7's transcriptional function and blocking its nuclear localization and DNA binding[40]. Based on our experimental findings, we suggest that therapeutic agents targeting the ANKRD10-2-MYC interaction pathway could offer a promising strategy for tumor suppression.

Finally, we conducted knockdown experiments targeting *ANKRD10-2* in *RBPMS* knockout cell lines. The results revealed that depletion of *ANKRD10*-2 led to decreased migration of tumor cells and decreased the expression levels of MYC target genes. This result confirmed the involvement of ANKRD10-2 in the progression of BLCA. Additionally, our investigation revealed that *ANKRD10-2* knockdown mitigated the effect of *RBPMS* knockout on tumor cell migration. This observation suggests that RBPMS exerts its inhibitory influence on BLCA metastasis by modulating mRNA splicing of *ANKRD10*.

We acknowledge certain methodological limitations in our study. In our experimental design, we employed two distinct sgRNAs for *RBPMS* knockout in bladder cancer cells and conducted comparative RNA-Seq analyses with wild-type cells. However, we observed limited concordance in RNA splicing events between the two knockout populations. This discrepancy may be attributed to two primary factors: potential off-target effects associated with the sgRNA methodology, and the accumulation of genetic variations during the extensive cell passaging required for single-clone expansion of knockout populations, which may have introduced cellular heterogeneity. For future studies, utilizing multiple siRNAs for transient transfection could enhance the consistency of experimental results. The animal experiments employed the tail vein lung metastasis model, which has limitations for studying bladder cancer cell migration in vivo. This model involves direct injection of tumor cells into the tail vein and cannot fully replicate natural metastasis. Specifically, it bypasses critical steps like tumor cell detachment from the primary site and blood vessel entry.

## Methods
### Cell culture
The cell lines used in this research were obtained from the Cell Bank of the Chinese Academy of Science (Shanghai, China) and validated using the short tandem repeat method. T24 and 5637 cells were cultured in RPMI 1640 (Gibco). J82 and UM-UC-3 cells were cultured in MEM (Gibco). RT4 cells were cultured in McCoy's 5 A medium (Gibco). HEK293T cells were cultured in Dulbecco's Modified Eagle Medium (DMEM, Hyclone). All media were supplemented with 10% fetal bovine serum (FBS, Hyclone).

### siRNAs and plasmids
The siRNAs and sgRNAs were purchased from GenePharma (Shanghai, China). Sequence information of the siRNAs and sgRNAs is presented in Supplementary Table 1.

The RBPMS-A, RBPMS-C, Flag-RBPMS-A, Flag-RBPMS-C, HA-ESRP1, Flag-ANKRD10-1 and Flag-ANKRD10-2 plasmids were purchased from GeneCopoeia (Guangzhou, China). The 5× E-box luciferase

plasmid was constructed and validated using molecular cloning techniques described in our previous study[41]. The procedure involved inserting five repeats of the E-box sequence (CACGTG) into the multiple cloning site of the pMCS-Fluc-SV40-hRluc vector using standard subcloning techniques. The HA-MYC plasmid was generously gifted by Professor Guoliang Qing of Wuhan University. Transfection of siRNA and plasmids was performed using LipofectamineTM 3000 Reagent (L3000075, Invitrogen), following the manufacturer's instructions.

### Antibodies
Information regarding the sources and dilution ratios of the antibodies is presented in Supplementary Table 2.

### Transwell assays
The transwell chamber was placed in a 24-well plate containing 600 μL of serum-containing medium. Next, $3 \times 10^4$ cells were mixed with 200 μL of serum-free medium and introduced into the transwell chamber. After 24 h of incubation, the bottom of transwell chamber was placed in 4% paraformaldehyde solution for 3 h. Upon completion of the fixation process, the cells underwent staining with a 0.1% crystal violet solution for an additional 3 h. The concluding steps involved rinsing the cells with distilled water, followed by drying and photographic documentation.

### Immunohistochemical (IHC) analysis
Tumor samples were obtained from nude mice, which are a model of tumor lung metastasis. Tissue microarray (HBlaU079Su01) was purchased from Shanghai Outdo Biotech Co., Ltd. The samples were fixed in formalin, dehydrated, and embedded in paraffin. Subsequently, sections were meticulously prepared, followed by deparaffinization, antigen retrieval, and blocking of the endogenous peroxidase activity. Following incubation with specific antibodies, the slides received a fresh addition of DAB staining solution. Ultimately, imaging of the slides was conducted utilizing a pathology slide scanner (Aperio VERSA 8, Leica).

### Wound healing assays
Upon reaching complete confluence in petri dish, cells were manually scratched, followed by immediate photographic documentation. Following a 24-hour incubation period, the previously scratched area was re-imaged. Subsequently, the captured images underwent analysis utilizing ImageJ software[42]. The gap closure percentage was calculated as follows: Gap closure (%) = (0 h area − 24 h area)/0 h area × 100%.

### RNA isolation and quantitative reverse-transcription PCR (qRT-PCR)
RNA extraction, cDNA synthesis, and qRT-PCR were performed as described previously[43]. Total RNA extraction was conducted utilizing the HiPure Total RNA Mini Kit (R4111-03, Magen). RNA concentration was subsequently quantified using a NanoDrop® ND-2000 UV-Vis spectrophotometer (Thermo Scientific, USA). Following the manufacturer's protocol, cDNA synthesis was performed using the ReverTra Ace qPCR RT Kit (FSQ-101, Toyobo). The quantitative analysis was carried out via qRT-PCR using iTaq Universal SYBR Green supermix (1725124, Bio-Rad). The primer sequences used for qRT-PCR are presented in Supplementary Table 3.

### Reverse transcription-PCR (RT-PCR) analysis
Following preparation of the reaction solution comprising template cDNA, upstream and downstream primers, and 2× TSINGKE Master Mix (TSE004, Tsingke) in a PCR tube, a brief centrifugation step was performed to ensure thorough mixing. The reaction was initiated using a PCR instrument. The resulting PCR products were visualized by agarose gel electrophoresis coupled with chemiluminescence detection. Gel images were captured using a gel imager (BioSpectrum 515 Imaging System, UVP). The primer sequences used for RT-PCR are presented in Supplementary Table 4.

## Animal models of lung metastasis

Lentiviruses expressing RBPMS-A or RBPMS-C were acquired from GenePharma (Shanghai, China) and were subsequently used for infection in T24 cells. After infection, T24 cells stably overexpressing RBPMS-A or RBPMS-C were subjected to puromycin selection (1 µg/mL, Sigma). Male BALB/c-nude mice were obtained from Beiente BioTechnology (Wuhan, China) and randomly divided into three groups: Vector group ($n = 6$), RBPMS-A overexpression group ($n = 6$), and RBPMS-C overexpression group ($n = 6$). All nude mice were housed in the specific-pathogen-free (SPF) facility at the Animal Experimental Center of Zhongnan Hospital of Wuhan University. No inclusion or exclusion criteria were established for the animals. The nude mice purchased were all 4 weeks old. Following one week of acclimatization feeding, $1 \times 10^6$ corresponding cells were injected into the tails of nude mice in each group. For the treatment, no blinding was done. After a six-week period, fluorescence detection was performed to analyze the nude mice. The mice were sacrificed by cervical dislocation. Subsequently, lung tissues were carefully isolated, photographed, and examined using H&E- and IHC-staining techniques. We have complied with all relevant ethical regulations for animal testing.

## RNA immunoprecipitation (RIP)

Specific experimental procedures were performed using the Pure-Binding® RNA Immunoprecipitation Kit (Geneseed, China). The magnetic beads were pre-treated and incubated with antibodies. Subsequently, $1 \times 10^7$ pre-treated cells were harvested, and 1 ml of lysate prepared using the kit was used for cell lysis on ice for 10 min. The cell lysate was then centrifuged at $10,000 \times g$ for 10 min at 4 °C. The resulting supernatant was transferred to RNase-free centrifuge tubes, with 100 µL of the supernatant allocated as the input group and stored at −20 °C, while the remaining supernatant was added to the antibody-bead complex and rotationally incubated for 12 h at 4 °C. The supernatant was discarded and the complex bound to the magnetic beads was eluted, facilitating the extraction of RNA along with the input group. Finally, RNA was detected using qRT-PCR.

## RNA-Sequencing (RNA-seq) and AS analyses

RNA was extracted from tumor cells using TRIzol reagent and processed using the KAPA Stranded mRNA-Seq Kit (Roche, USA), following the manufacturer's instructions. RNA fragments with polyA tails were selectively captured using oligo-dT beads. Subsequently, the captured mRNA fragments were denatured thermally, resulting in fragmentation of 200–300 bp. Following fragmentation, the fragments were incubated with a strand synthesis master mix to convert them to cDNA. The resulting cDNA molecules were then end-repaired, followed by the addition of an adenine (A) base at the 3' end, and subsequent ligation into sequencing junctions. PCR amplification was used to generate libraries for high-throughput sequencing, with DNA fragments typically spanning 300–400 bp in length. The libraries were thoroughly examined, tested, and sequenced using the Illumina NovaSeq 6000 platform. Following sequencing, we obtained raw data (raw reads) and used Cutadapt to remove adapter sequences from the original data[44]. We then used Trimmomatic to filter out low-quality sequences, generating clean data. FastQC was used to analyze the clean data volume and calculate q20 and q30 proportions[45]. Next, HISAT2 aligned the clean data to the reference genome[46]. Finally, StringTie assembled the reads into transcripts and analyzed gene expression levels[47]. Differential gene analysis was conducted using the R package "Deseq2," while functional enrichment analyses, including GO, KEGG and GSEA, were performed using the R package "clusterProfiler." The Replicate Multivariate Analysis of Transcript Splicing (rMATS) software was developed to analyze alternative splicing events between replicate transcriptome samples[48]. By comparing AS differences between two samples or groups of samples (with replicates in each group), it generates statistical results for common and significantly different splicing events. Alternative splicing events with | IncLevelDifference| > 0.1 and $p$-value < 0.05 were considered significant, with results visualized using the "ggplot2" package in R.

## Luciferase reporter assays

293 T cells were transfected with luciferase plasmid when grown to the appropriate density. After 8 h other plasmids or siRNAs were transfected into cells and the culture continues for 48 h. Following the manufacturer's instructions, the specific experimental procedure was performed using the Dual-luciferase Reporter Assay System kit (E1910, Promega).

## Western blot analysis

Lysates were prepared using RIPA buffer, along with protease and phosphatase inhibitors, then added to centrifugally collected cells, followed by mixing for lysis on ice. After a 30-min lysis period, the lysate was combined with 5× SDS-PAGE loading buffer and denatured at 100 °C. Following the separation of total protein through electrophoresis, it was transferred onto a membrane made of polyvinylidene difluoride (PVDF). Subsequently, the PVDF membrane underwent blocking using a solution containing 5% skim milk. Then, it was subjected to an incubation process with specific primary antibodies, succeeded by a subsequent incubation with corresponding secondary antibodies. Eventually, the visualization of protein bands occurred through chemiluminescence detection, and their documentation was executed employing a gel imager (BioSpectrum 515 Imaging System, UVP).

## Co-immunoprecipitation (Co-IP) assays

According to the BeaverBeads™ Protein A/G Immunoprecipitation Kit protocol (Beaver, China), Co-IP assays were conducted. Lysates were prepared using IP binding buffer, along with protease and phosphatase inhibitors, then added to centrifugally collected cells, followed by mixing for lysis on ice. After a 30-min lysis period, the cell lysate was then centrifuged at $10,000 \times g$ for 10 min at 4 °C. The supernatant was divided into input and IP samples. The input sample was combined with 5× SDS-PAGE loading buffer, denatured by heating at 100 °C and stored at −20 °C. Appropriate antibodies were added to the IP samples, which were then rotationally incubated for 12 h at 4 °C. The magnetic beads were washed 3 times with IP binding buffer and added to the antibody-protein complex. The supernatant was discarded after 2 h. The magnetic beads were then washed 3 times again and added to 1× SDS-PAGE loading buffer. Finally, IP samples were denatured by heating at 100 °C and analyzed via Western blotting alongside input samples.

## Chromatin immunoprecipitation (ChIP)

The ChIP analyses were conducted in accordance with the methodology outlined in the Pierce Magnetic ChIP Kit (#26157, Thermo Fisher Scientific). The protocol begins with protein-DNA crosslinking through a 10-min treatment of cells with 1% formaldehyde, followed by crosslink termination using 0.125 mM glycine for 5 min. Subsequently, cells undergo three PBS wash cycles before nuclear isolation is achieved using membrane extraction buffer supplemented with protease/phosphatase inhibitors. Chromatin digestion is performed using Micrococcal nuclease (MNase) at 37 °C for 10 min, after which the reaction is halted with MNase Stop Solution. The nuclear fraction is then reconstituted in 500 µL of 1× IP dilution buffer. Nuclear membrane disruption is accomplished using a Q800R3 non-contact sonicator (Qsonica). A 5 µL aliquot of the sonicated chromatin is reserved as Input, while the remainder undergoes immunoprecipitation following mixture with 1× IP dilution buffer and the addition of IgG and MYC target antibodies. This mixture is subjected to overnight incubation at 4 °C with continuous rotation.

The immunoprecipitation process continues with the addition of 20 µL ChIP-grade protein A/G magnetic beads, followed by a 2-h incubation at 4 °C and subsequent magnetic separation. The protocol proceeds with sequential washing steps utilizing low-salt and high-salt IP wash buffers, followed by elution in 150 µL of 1× IP elution buffer at

**Article**

65 °C for 30 min. Crosslink reversal is achieved through a 4-hour incubation at 65 °C with 6 µL of 5 M NaCl and 2 µL of 20 mg/mL proteinase K. The final phase involves DNA purification via specialized columns, incorporating binding and washing steps, culminating in elution with 200 µL of elution buffer. The purified DNA specimens are then suitable for qPCR analysis. The specific primer sequences are detailed in Supplementary Table 5.

### Crosslinking immunoprecipitation (CLIP)

We conducted the CLIP experiments following the protocol provided in the BersinBio CLIP-qPCR Kit (#Bes3014-1, Bersinbio). 5637 cells at 80% confluence were treated with 100 µM 4-thiouridine for 16 h and washed once with pre-chilled PBS. The cells were placed on ice and crosslinked using a Spectrolinker XL-1500 (SpectronicsCorporation) at 365 nm wavelength ($0.15 J/cm^2$) for 10 min. Cells were then scraped in PBS, centrifuged, and the supernatant was removed. The cells were lysed on ice for 10 min in lysis buffer containing DTT and Protease inhibitor, then centrifuged at 13,000 g for 15 min at 4 °C. The supernatant was treated with RNase T1 (final concentration 1 U/µL) at 22 °C for 15 min, then immediately chilled on ice for 5 min. A 1% aliquot of supernatant was saved as Input and stored at −80 °C, while the remaining solution was split and incubated overnight at 4 °C with IgG and RBPMS antibodies on a vertical mixer. Magnetic beads were added to both IP and IgG groups and incubated for 3 h. After washing with IP buffer and magnetic separation, the samples were treated with 100 µL IP wash buffer and 20 µL DNase I, incubated in a 37 °C water bath for 15 min, then immediately chilled on ice for 5 min. Proteinase K digestion was performed at 55 °C for 30 min, followed by magnetic separation and collection of the supernatant in RNase-free tubes. RNA was extracted by adding 0.8 mL Trizol and 160 µL chloroform, followed by centrifugation at $12,000 \times g$ for 10 min at 4 °C. The RNA was precipitated overnight at −80 °C with isopropanol, centrifuged, washed with ethanol, and reconstituted in 20 µL $ddH_2O$ after drying. For 3' adapter ligation and reverse transcription, a 20 µL reaction mixture was prepared containing 1 µL Poly A Polymerase, 1 µL RTaseMix, 4 µL 5×PAP/RT Buffer, 10 µL template RNA, and $ddH_2O$. The reaction was run at 37 °C for 1 h followed by 85 °C for 5 min. The resulting cDNA underwent qPCR analysis using a 20 µL reaction mixture containing 10 µL 2×SYBR Master Mix, 1 µL each of forward and reverse primers (10 µM), 2 µL cDNA template, and $ddH_2O$. The specific primer sequences are detailed in Supplementary Table 5.

### Statistics and reproducibility

Bioinformatics analysis of the data was conducted using R (version 4.1.3) software, and GraphPad Prism software was used for statistical analysis of the experimental data. The log-rank test was used for statistical analysis of survival data. A two-tailed unpaired Student's $t$ test was employed for statistical analysis of two data groups. For data from multiple groups, one-way ANOVA with Dunnett's multiple comparison test was utilized. Data are presented as the mean ± standard deviation (SD). For GO and GSEA analyses in RNA-seq, statistical significance was assessed using a two-tailed Fisher's exact test. Cell counting and quantification of protein band gray values were conducted using ImageJ software[49].

### Reporting summary

Further information on research design is available in the Nature Portfolio Reporting Summary linked to this article.

### Data availability

The RNA-seq data generated in this study are publicly available in the Gene Expression Omnibus (GEO) datasets GSE267762. TCGA-BLCA data were downloaded from the UCSC Xena database. GSE13507, GSE32548, GSE32894, GSE3167, GSE48075, GSE83596, and GSE120736 datasets were downloaded from the GEO database. Summary data are presented in the paper and Supplementary Files. The supplementary information document contains all Supplementary Figs. (Supplementary Fig. 1–5), original uncropped protein immunoblots (Supplementary Fig. 6), and original

uncropped agarose gels (Supplementary Fig. 7). Supplementary Data 1 presents the alternative splicing events identified through RNA-Seq analysis after RBPMS knockout. Supplementary Data 2 contains the source data. All other data are available from the corresponding author on reasonable request.

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

## Acknowledgements

The authors acknowledge Seon-Kyu Kim, David Lindgren, Gottfrid Sjodahl, Lars Dyrskjot, Woonyoung Choi, Aline de Conti and Bic-Na Song for making their RNA-seq data publicly available. This study was supported by National Natural Science Foundation of China (Grant No. 82103609).

## Author contributions

J.Y., H.Z., and M.L. were responsible for designing the study and writing the manuscript. J.Y., L.C., and M.L. contributed to the cellular and biochemical experiments. J.Y., G.W., and K.Q. performed the animal experiments. L.C., G.W., H.W., and Z.Y. helped with data curation and assembly. J.Y., H.Z., and M.L. performed data analysis and visualization. H.Z. and M.L. were responsible for funding acquisition and supervision. All authors corrected the final manuscript.

## Competing interests

The authors declare no competing interests.

## Ethical approval

The animal study was approved by the Experimental Animal Welfare Ethics Committee, Zhongnan Hospital of Wuhan University (approval number: ZN2022271). We have complied with all relevant ethical regulations for animal testing.
