## [Transparent Peer Review file · Communications Biology]

RBPMS inhibits bladder cancer metastasis by downregulating MYC pathway through alternative splicing of ANKRD10

Corresponding Author: Dr Mengxin Lu

Version 0:

Reviewer comments:

Reviewer #1

(Remarks to the Author)

The manuscript entitled “RBPMS inhibits bladder cancer migration and invasion by downregulating the MYC pathway through alternative splicing of ANKRD10” investigates the role of RBPMS, an RNA-binding protein, in bladder cancer (BLCA) and its impact on metastasis. It provides substantial evidence that RBPMS downregulation is associated with BLCA progression, particularly in muscle-invasive bladder cancer (MIBC). The research effectively demonstrates that RBPMS inhibits BLCA cell migration and invasion both in vitro and in vivo by regulating the alternative splicing of ANKRD10 mRNA, which impacts MYC pathway activity. The manuscript is well-structured, and the experiments are robust and detailed. I have a few suggestions to improve the manuscript before publication.

1. The Materials and Methods section would benefit from more comprehensive details regarding the reagents used, including manufacturers' catalog numbers and specific concentrations. Additionally, more detailed descriptions of experimental protocols, particularly for cell culture conditions, RNA sequencing, and the methods used for mRNA splicing analysis, should be included. Clarifying these steps will enhance reproducibility and transparency.
2. Figure 1i is not described in the manuscript. The authors should ensure that all figures are accurately referenced and described in the main text.
3. The authors should include the uncropped and triplicate original images of Western blots as supplementary materials. Providing these images would increase the transparency of the experimental results and allow for a more thorough evaluation of the data.
4. There are discrepancies in the detection of N-cadherin and Vimentin proteins between Figure 2d (vector group) and Supplementary Figure 3c (sgCtrl group). Additionally, the Actin control shows two bands in some images but only one in others. The authors should address these inconsistencies, possibly by re-evaluating the experimental conditions, ensuring standardized protein detection methods, and clarifying any technical differences that might explain these observations.
5. The RNA-seq data in Figure 3 reveals considerable differences between the sgRBPMS-1 and sgRBPMS-2 cell lines. The authors should explore potential reasons for these discrepancies, such as off-target effects of the knockout strategy or biological variability between cell lines. Addressing this point could clarify the robustness of the RNA-seq findings and the specificity of the RBPMS knockout?
6. While the study provides a novel insight into the regulatory role of RBPMS in BLCA, the potential clinical relevance of targeting RBPMS or ANKRD10 for therapeutic purposes could be further discussed. Expanding on the potential translational implications would strengthen the manuscript and highlight the broader significance of the findings.
7. The mechanistic model presented is well-supported by the data; however, the direct interaction between ANKRD10-2 and MYC could be investigated further to validate their co-activator relationship. Co-immunoprecipitation (Co-IP) experiments were informative, but additional assays, such as chromatin immunoprecipitation (ChIP), could be employed to demonstrate the recruitment of MYC and ANKRD10-2 to target gene promoters, thus reinforcing the proposed regulatory pathway.
8. The Discussion section could benefit from a more explicit acknowledgment of the study's limitations, particularly concerning the potential off-target effects of siRNA and overexpression systems used in vitro. A critical assessment of these limitations would provide a balanced perspective on the findings and suggest directions for future research.

Reviewer #2

(Remarks to the Author)

Yu and colleagues focus on genes differentially expressed between carcinoma and adjacent non-cancerous tissues in bladder cancer (BLCA) to identify the most significant differences in high or low expression between cancer and paracancer tissues of the TCGA-BLCA cohort.

They identify 5 RNA-binding proteins. Among them, the authors decide to focus on the RNA-binding protein mRNA processing factor (RBPMS), which was downregulated in BLCA patients. Remarkably, elevated RBPMS expression was associated with improved prognosis for BLCA patients.

The authors show that RBPMS impacts on cell migration and invasion and also exerts inhibitory effects on the epithelial mesenchymal transition (EMT) pathway. RNA sequencing experiments revealed that RBPMS depletion in BLCA cells resulted in several alternative spliced transcripts, including the ANKRD10, with increased expression of the ANKRD10-2 isoform.

ANKRD10-2 functioned as a transcriptional co-activator of MYC proteins

The authors conclude that RBPMS suppresses the migration and invasion of BLCA cells by attenuating the MYC pathway through regulation of ANKRD10 splicing.

The study is potentially interesting, but several key details were not provided, including information on the ANKRD10 splicing isoforms. Moreover, the list of the identified splicing events was not provided, and the impact of RBPMS on splicing was not addressed.

Additional points to carefully address are listed below.

1. The authors claim that "Following comprehensive functional inquiries into the cancer upregulated 500 genes in the GO database, they identified five differentially expressed RNA-binding proteins. They authors should report the list of the 500 genes as a new supplementary table and clarify how these 5 RBPs came up.
2. Bioinformatic analysis upon RBPMS knockdown is not very detailed.
3. Why do the authors decide to focus on ANKRD10?
4. The authors claim that RBPMS exerts an inhibitory effect on BLCA metastasis through the cleavage of ANKRD10, how do they reach this conclusion?
5. The authors performed RNA-seq analysis to identify transcripts alternatively spliced upon RBPMS knockdown. However, the list of genes and splicing events regulated is not provided. The lists should be provided as supplementary Tables.
6. How do the authors explain the low overlap between siRBPMS1 and siRBPMS2 in terms of splicing regulation? How does RBPMS1 work in splicing regulation?
7. CLIP experiments should be performed to verify the direct binding of RBPMS to ANKRD10.
8. The scheme of ANKRD10 splicing is not very clear. Moreover, since ANKRD10 splicing has not been described before, a more detailed characterization of these splice variants should be performed. What is the difference in term of protein domain, between the two splicing variants?
It would help the comprehension to indicate the ensembl id.
9. Densitometric analysis of the splicing changes should be added to Fig.3F.
10. Two bands are associated with the "inclusion" of ANKRD10 at 216bp (Fig.3f,l); did the authors sequence these bands? Which one did they clone for the IP experiments in Fig.5g)?

Version 1:

Reviewer comments:

Reviewer #1

(Remarks to the Author)

The authors have addressed my concerns.

Reviewer #2

(Remarks to the Author)

The authors have satisfactorily addressed most of my previous concerns.

MS No.: COMMSBIO-24-5112

Title: RBPMS inhibits bladder cancer migration and invasion by downregulating the MYC pathway through alternative splicing of ANKRD10

To whom it may concern,

Thank you for giving us a chance to revise our manuscript entitled “**RBPMS inhibits bladder cancer migration and invasion by downregulating the MYC pathway through alternative splicing of ANKRD10**” (COMMSBIO-24-5112). In the revised manuscript, we have added experimental data as suggested and carefully revised the manuscript, and provided the lists of genes and splicing events in the supplementary data. Below is our point-by-point response to each comment, in which all the text in blue are the original comments from the reviewers and our response in black.

Reviewer #1 (Comments to the Author):

The manuscript entitled “RBPMS inhibits bladder cancer migration and invasion by downregulating the MYC pathway through alternative splicing of ANKRD10” investigates the role of RBPMS, an RNA-binding protein, in bladder cancer (BLCA) and its impact on metastasis. It provides substantial evidence that RBPMS downregulation is associated with BLCA progression, particularly in muscle-invasive bladder cancer (MIBC). The research effectively demonstrates that RBPMS inhibits BLCA cell migration and invasion both in vitro and in vivo by regulating the alternative splicing of ANKRD10 mRNA, which impacts MYC pathway activity. The manuscript is well-structured, and the experiments are robust and detailed. I have a few suggestions to improve the manuscript before publication.

Re: We sincerely appreciate your thorough review and recognition of our manuscript. Your summary accurately captures the core findings of our research. It is encouraging that our work in elucidating the role of RBPMS as an RNA binding protein in bladder cancer, particularly in regulating MYC pathway and tumor metastasis through ANKRD10 alternative splicing, has been fully acknowledged. Your positive assessment reinforces our belief in the significance of this study. We are very grateful for your recognition of our experimental rigor and manuscript structure. We will carefully

consider all your suggestions to further improve our work.

1. The Materials and Methods section would benefit from more comprehensive details regarding the reagents used, including manufacturers' catalog numbers and specific concentrations. Additionally, more detailed descriptions of experimental protocols, particularly for cell culture conditions, RNA sequencing, and the methods used for mRNA splicing analysis, should be included. Clarifying these steps will enhance reproducibility and transparency.

Re: Thank you for your suggestions. We have enhanced the Materials and Methods section by including detailed manufacturer catalog numbers and specific concentrations (line 107-110, yellow background mark). We have also supplemented detailed descriptions of our cell culture conditions, RNA sequencing protocols, and mRNA splicing analysis methods (line 208-223, yellow background mark).

2. Figure 1i is not described in the manuscript. The authors should ensure that all figures are accurately referenced and described in the main text.

Re: We appreciate you identifying this matter. Upon review, we noted that while Figure 1i was appropriately described within the main text, there was an inconsistency in the figure citation at the conclusion, which has since been amended to accurately reference Figure 1i (line 381, yellow background mark).

3. The authors should include the uncropped and triplicate original images of Western blots as supplementary materials. Providing these images would increase the transparency of the experimental results and allow for a more thorough evaluation of the data.

Re: We appreciate your feedback regarding the Western blot documentation. As requested, we have incorporated the unmodified Western blot images into Supplementary Figure S6 for comprehensive validation of our findings.

4. There are discrepancies in the detection of N-cadherin and Vimentin proteins between Figure 2d (vector group) and Supplementary Figure 3c (sgCtrl group). Additionally, the Actin control shows two bands in some images but only one in others. The authors should address these inconsistencies, possibly by re-evaluating the experimental conditions, ensuring standardized protein detection methods, and clarifying any technical differences that might explain these observations.

Re: Thank you for your questions regarding our experiments. Our Western blot analyses were performed following standardized protocols across all experiments. The observed variations in N-cadherin and Vimentin protein levels between Figure 2d (vector group) and Supplementary Figure 3c (sgCtrl group) reflect the distinct molecular characteristics of the T24 and 5637 bladder cancer cell lines, which originate from different patients. To comprehensively evaluate RBPMS's regulatory role in the EMT pathway, we strategically selected the 5637 cell line, which exhibits elevated RBPMS expression, for knockdown studies, and the T24 cell line, which demonstrates lower RBPMS expression, for overexpression analyses. In our Western blot results, Actin consistently appears as double bands. This pattern occurs because we used high-concentration SDS-PAGE gels for protein electrophoresis to resolve RBPMS, which is a small protein. The high gel concentration led to enhanced separation of the Actin protein bands.

5. The RNA-seq data in Figure 3 reveals considerable differences between the sgRBPMS-1 and sgRBPMS-2 cell lines. The authors should explore potential reasons for these discrepancies, such as off-target effects of the knockout strategy or biological variability between cell lines. Addressing this point could clarify the robustness of the RNA-seq findings and the specificity of the RBPMS knockout?

Re: We appreciate your thoughtful observation. To ensure the robustness of our findings and minimize potential experimental bias, we implemented a dual-knockout strategy using two independent RBPMS knockout cell lines for our RNA-Seq analysis. This methodological approach enabled us to more accurately identify and validate the splicing events under RBPMS regulation. We have thoroughly addressed these experimental considerations in the results and discussion sections (line 426-427 and line 634-643, yellow background mark).

6. While the study provides a novel insight into the regulatory role of RBPMS in BLCA, the potential clinical relevance of targeting RBPMS or ANKRD10 for therapeutic purposes could be further discussed. Expanding on the potential translational implications would strengthen the manuscript and highlight the broader significance of the findings.

Re: We appreciate your valuable suggestion. After reviewing the relevant literature, we have incorporated a comprehensive discussion of the potential clinical implications of

RBPMS and ANKRD10-targeted therapies in the discussion section (line 614-625, yellow background mark).

7. The mechanistic model presented is well-supported by the data; however, the direct interaction between ANKRD10-2 and MYC could be investigated further to validate their co-activator relationship. Co-immunoprecipitation (Co-IP) experiments were informative, but additional assays, such as chromatin immunoprecipitation (ChIP), could be employed to demonstrate the recruitment of MYC and ANKRD10-2 to target gene promoters, thus reinforcing the proposed regulatory pathway.

Re: Following your suggestion, we conducted ChIP-qPCR experiments (line 260-284, yellow background mark). Based on our previous experimental results showing high MYC protein expression in 5637 cells, we knocked down ANKRD10-1 and ANKRD10-2 separately in 5637 cells, performed enrichment using IgG and MYC antibodies followed by qPCR detection. The results showed that after ANKRD10-2 knockdown, MYC binding to target genes was significantly weakened, while ANKRD10-1 depletion demonstrated minimal effect on MYC's association with target gene promoters (line 537-543, yellow background mark). The specific experimental results are presented in Figure 5i.

8. The Discussion section could benefit from a more explicit acknowledgment of the study's limitations, particularly concerning the potential off-target effects of siRNA and overexpression systems used in vitro. A critical assessment of these limitations would provide a balanced perspective on the findings and suggest directions for future research

Re: We appreciate your valuable feedback regarding study limitations. In response, we have conducted a thorough analysis of potential constraints in our research methodology and incorporated these insights into the discussion section to provide a more comprehensive perspective (line 634-643, yellow background mark).

Reviewer #2 (Comments to the Author):

Yu and colleagues focus on genes differentially expressed between carcinoma and adjacent non-cancerous tissues in bladder cancer (BLCA) to identify the most significant differences in high or low expression between cancer and paracancer tissues

of the TCGA-BLCA cohort.

They identify 5 RNA-binding proteins. Among them, the authors decide to focus on the RNA-binding protein mRNA processing factor (RBPMS), which was downregulated in BLCA patients. Remarkably, elevated RBPMS expression was associated with improved prognosis for BLCA patients.

The authors show that RBPMS impacts on cell migration and invasion and also exerts inhibitory effects on the epithelial mesenchymal transition (EMT) pathway. RNA sequencing experiments revealed that RBPMS depletion in BLCA cells resulted in several alternative spliced transcripts, including the ANKRD10, with increased expression of the ANKRD10-2 isoform.

ANKRD10-2 functioned as a transcriptional co-activator of MYC proteins

The authors conclude that RBPMS suppresses the migration and invasion of BLCA cells by attenuating the MYC pathway through regulation of ANKRD10 splicing.

The study is potentially interesting, but several key details were not provided, including information on the ANKRD10 splicing isoforms. Moreover, the list of the identified splicing events was not provided, and the impact of RBPMS on splicing was not addressed.

Additional points to carefully address are listed below.

Re: Thank you for your thorough review and thoughtful recognition of our work. Your summary effectively captures our study's key findings—from identifying differentially expressed genes in BLCA tissues, to uncovering RBPMS downregulation in BLCA patients and its prognostic implications, to explaining how RBPMS affects cell migration, invasion, and the EMT pathway. We are especially encouraged by your acknowledgment of our discovery that RBPMS inhibits the MYC pathway by regulating ANKRD10 splicing, which influences BLCA cell migration and invasion. We understand your concerns about the need for additional data on ANKRD10 splice isoforms and RBPMS's splicing effects. We will include this essential information in our revised manuscript. Your feedback will help strengthen our study's scientific rigor and completeness. We will address each of your revision suggestions to enhance this work.

1. The authors claim that “Following comprehensive functional inquiries into the cancer upregulated 500 genes in the GO database, they identified five differentially expressed RNA-binding proteins. They authors should report the list of the 500 genes

as a new supplementary table and clarify how these 5 RBPs came up.

Re: Thank you for your suggestion. We have included the gene list in Supplementary Table S6. We input these 500 genes into the GO database (<https://geneontology.org/>) and searched the pathway enrichment results for genes associated with the RNA binding term (GO:0003723). This analysis identified five RNA binding-related genes that we selected for further investigation. We have added a detailed description of this analytical process to the “Results” section (line 328-338, yellow background mark).

2. Bioinformatic analysis upon RBPMS knockdown is not very detailed.

Re: We appreciate your valuable feedback. In response, we conducted KEGG pathway enrichment analysis on the differentially expressed genes identified after RBPMS knockout, with the findings illustrated in Figure S4b. Additionally, we have enhanced the Materials and Methods section with a comprehensive description of our bioinformatics analytical methodology (line 208-223, yellow background mark).

3. Why do the authors decide to focus on ANKRD10?

Re: To identify genes and splicing events regulated by RBPMS, we employed a systematic approach utilizing dual sgRNA-mediated knockdown followed by comprehensive transcriptome and alternative splicing analysis. Our validation studies, which included PCR experiments comparing wild-type and RBPMS knockout cell lines, revealed that ANKRD10 exhibited the most pronounced alternative splicing response to both RBPMS depletion and overexpression. Based on these compelling findings, we selected ANKRD10 as the focus of our subsequent investigations (line 442-444, yellow background mark).

4. The authors claim that RBPMS exerts an inhibitory effect on BLCA metastasis through the cleavage of ANKRD10, how do they reach this conclusion?

Re: Our research demonstrates that RBPMS knockdown leads to enhanced bladder cancer cell migration, as evidenced by comprehensive in vitro and in vivo analyses. Through RNA-Seq analysis and subsequent biochemical validation experiments, we identified ANKRD10 as the primary alternative splicing target regulated by RBPMS. Specifically, we observed a notable increase in ANKRD10-2 splice variant expression following RBPMS depletion. Further investigation revealed that ANKRD10-2 functions as a binding partner of the MYC oncoprotein, facilitating its regulatory

activity on target genes. To establish causality, we performed rescue experiments and found that ANKRD10-2 knockdown partially reversed the enhanced migratory phenotype induced by RBPMS depletion. These experimental findings provide strong evidence that RBPMS regulates BLCA metastasis through its modulatory effect on ANKRD10 splicing.

5. The authors performed RNA-seq analysis to identify transcripts alternatively spliced upon RBPMS knockdown. However, the list of genes and splicing events regulated is not provided. The lists should be provided as supplementary Tables.

Re: Thank you for your suggestion. We have included a comprehensive list of regulated genes and splicing events identified from the RNA-seq data following RBPMS knockout in Supplementary Data 1.

6. How do the authors explain the low overlap between siRBPMS1 and siRBPMS2 in terms of splicing regulation? How does RBPMS1 work in splicing regulation?

Re: We appreciate your thoughtful observation regarding the overlap between sgRBPMS1 and sgRBPMS2. To ensure the robustness of our findings and minimize potential experimental bias, we implemented a dual-knockdown strategy using two independent sgRNA constructs. This methodological approach enabled us to more accurately identify and validate the splicing events under RBPMS regulation. We have thoroughly addressed these experimental considerations in the results and discussion sections (line 426-427, line 634-643, yellow background mark).

7. CLIP experiments should be performed to verify the direct binding of RBPMS to ANKRD10.

Re: In response to your constructive feedback, we performed CLIP-qPCR analysis to examine the molecular interaction between RBPMS and ANKRD10 RNA. Our experimental findings demonstrated a direct binding relationship, with the results documented in Figure 3K (line 286-313, line 454-460, yellow background mark).

8. The scheme of ANKRD10 splicing is not very clear. Moreover, since ANKRD10 splicing has not been described before, a more detailed characterization of these splice variants should be performed. What is the difference in term of protein domain, between the two splicing variants?

It would help the comprehension to indicate the ensembl id.

Re: In response to your feedback, we have enhanced the schematic representation of ANKRD10 splicing patterns and indicated the ensembl id (Figure 3g). Our analysis reveals that both splice variants maintain identical sequences across the initial three exons, specifically in amino acid positions 1-151, which encompasses the conserved ANK repeat domain. Beyond this conserved region, the variants diverge significantly in their exon composition, leading to distinct amino acid sequences from position 152 onward. Notably, ANKRD10-2 exhibits a more compact sequence architecture and consequently a reduced molecular mass compared to ANKRD10-1, as evidenced by its lower migration pattern in Western blot analysis (Figure 3j).

9. Densitometric analysis of the splicing changes should be added to Fig.3F.

Re: In response to this methodological suggestion, we conducted a quantitative analysis of PCR band intensity utilizing ImageJ software. The analysis incorporated β -actin as an internal reference standard for band normalization. To enhance data visualization, we have presented these quantitative findings as a comprehensive heat map in Figure 3f, effectively illustrating the regulatory impact of RBPMS on ANKRD10 splicing patterns (line 442-444, yellow background mark).

10. Two bands are associated with the "inclusion" of ANKRD10 at 216bp (Fig.3f,l); did the authors sequence these bands? Which one did they clone for the IP experiments in Fig.5g)?

Re: We appreciate your thoughtful observation regarding the two bands in the "inclusion" of ANKRD10 at 216bp. We named the longer ANKRD10 transcript variant as ANKRD10-1 (NM_017664.4) and the shorter transcript variant as ANKRD10-2 (NM_001286721.3). It is worth noting that although ANKRD10-1 has longer nucleotide and amino acid sequences, due to the ANKRD10-1 splicing event spanning an exon, the PCR product of ANKRD10-1 is shorter than that of ANKRD10-2. We have added more descriptions and annotations in the manuscript and the splicing diagram of ANKRD10 (Figure 3g) to provide a clearer and more intuitive understanding of the experimental results.

As shown in the following picture, we reanalyzed the original results of RT-PCR analysis. By compared with other genes bands and DNA maker, the higher density bands are more consistent with the "inclusion" bands of ANKRD10 at 216bp (marked

with red arrow), which were used to analyze ANKRD10 splicing, but were not used to clone the plasmids. And the Flag-ANKRD10-1 and Flag-ANKRD10-2 plasmids for the IP experiments were purchased from GeneCopoeia (Guangzhou, China) after sequencing, and we have added this information in the Materials Methods section (line 116-117, yellow background mark).

MS No.: COMMSBIO-24-5112

Title: RBPMS inhibits bladder cancer migration and invasion by downregulating the MYC pathway through alternative splicing of ANKRD10

To whom it may concern,

Thank you for giving us a chance to revise our manuscript entitled “**RBPMS inhibits bladder cancer migration and invasion by downregulating the MYC pathway through alternative splicing of ANKRD10**” (COMMSBIO-24-5112). In the revised manuscript, we have added experimental data as suggested and carefully revised the manuscript, and provided the lists of genes and splicing events in the supplementary data. Below is our point-by-point response to each comment, in which all the text in blue are the original comments from the reviewers and our response in black.

Reviewer #1 (Comments to the Author):

The manuscript entitled “RBPMS inhibits bladder cancer migration and invasion by downregulating the MYC pathway through alternative splicing of ANKRD10” investigates the role of RBPMS, an RNA-binding protein, in bladder cancer (BLCA) and its impact on metastasis. It provides substantial evidence that RBPMS downregulation is associated with BLCA progression, particularly in muscle-invasive bladder cancer (MIBC). The research effectively demonstrates that RBPMS inhibits BLCA cell migration and invasion both in vitro and in vivo by regulating the alternative splicing of ANKRD10 mRNA, which impacts MYC pathway activity. The manuscript is well-structured, and the experiments are robust and detailed. I have a few suggestions to improve the manuscript before publication.

Re: We sincerely appreciate your thorough review and recognition of our manuscript. Your summary accurately captures the core findings of our research. It is encouraging that our work in elucidating the role of RBPMS as an RNA binding protein in bladder cancer, particularly in regulating MYC pathway and tumor metastasis through ANKRD10 alternative splicing, has been fully acknowledged. Your positive assessment reinforces our belief in the significance of this study. We are very grateful for your recognition of our experimental rigor and manuscript structure. We will carefully

consider all your suggestions to further improve our work.

1. The Materials and Methods section would benefit from more comprehensive details regarding the reagents used, including manufacturers' catalog numbers and specific concentrations. Additionally, more detailed descriptions of experimental protocols, particularly for cell culture conditions, RNA sequencing, and the methods used for mRNA splicing analysis, should be included. Clarifying these steps will enhance reproducibility and transparency.

Re: Thank you for your suggestions. We have enhanced the Materials and Methods section by including detailed manufacturer catalog numbers and specific concentrations (line 107-110, yellow background mark). We have also supplemented detailed descriptions of our cell culture conditions, RNA sequencing protocols, and mRNA splicing analysis methods (line 208-223, yellow background mark).

2. Figure 1i is not described in the manuscript. The authors should ensure that all figures are accurately referenced and described in the main text.

Re: We appreciate you identifying this matter. Upon review, we noted that while Figure 1i was appropriately described within the main text, there was an inconsistency in the figure citation at the conclusion, which has since been amended to accurately reference Figure 1i (line 381, yellow background mark).

3. The authors should include the uncropped and triplicate original images of Western blots as supplementary materials. Providing these images would increase the transparency of the experimental results and allow for a more thorough evaluation of the data.

Re: We appreciate your feedback regarding the Western blot documentation. As requested, we have incorporated the unmodified Western blot images into Supplementary Figure S6 for comprehensive validation of our findings.

4. There are discrepancies in the detection of N-cadherin and Vimentin proteins between Figure 2d (vector group) and Supplementary Figure 3c (sgCtrl group). Additionally, the Actin control shows two bands in some images but only one in others. The authors should address these inconsistencies, possibly by re-evaluating the experimental conditions, ensuring standardized protein detection methods, and clarifying any technical differences that might explain these observations.

Re: Thank you for your questions regarding our experiments. Our Western blot analyses were performed following standardized protocols across all experiments. The observed variations in N-cadherin and Vimentin protein levels between Figure 2d (vector group) and Supplementary Figure 3c (sgCtrl group) reflect the distinct molecular characteristics of the T24 and 5637 bladder cancer cell lines, which originate from different patients. To comprehensively evaluate RBPMS's regulatory role in the EMT pathway, we strategically selected the 5637 cell line, which exhibits elevated RBPMS expression, for knockdown studies, and the T24 cell line, which demonstrates lower RBPMS expression, for overexpression analyses. In our Western blot results, Actin consistently appears as double bands. This pattern occurs because we used high-concentration SDS-PAGE gels for protein electrophoresis to resolve RBPMS, which is a small protein. The high gel concentration led to enhanced separation of the Actin protein bands.

5. The RNA-seq data in Figure 3 reveals considerable differences between the sgRBPMS-1 and sgRBPMS-2 cell lines. The authors should explore potential reasons for these discrepancies, such as off-target effects of the knockout strategy or biological variability between cell lines. Addressing this point could clarify the robustness of the RNA-seq findings and the specificity of the RBPMS knockout?

Re: We appreciate your thoughtful observation. To ensure the robustness of our findings and minimize potential experimental bias, we implemented a dual-knockout strategy using two independent RBPMS knockout cell lines for our RNA-Seq analysis. This methodological approach enabled us to more accurately identify and validate the splicing events under RBPMS regulation. We have thoroughly addressed these experimental considerations in the results and discussion sections (line 426-427 and line 634-643, yellow background mark).

6. While the study provides a novel insight into the regulatory role of RBPMS in BLCA, the potential clinical relevance of targeting RBPMS or ANKRD10 for therapeutic purposes could be further discussed. Expanding on the potential translational implications would strengthen the manuscript and highlight the broader significance of the findings.

Re: We appreciate your valuable suggestion. After reviewing the relevant literature, we have incorporated a comprehensive discussion of the potential clinical implications of

RBPMS and ANKRD10-targeted therapies in the discussion section (line 614-625, yellow background mark).

7. The mechanistic model presented is well-supported by the data; however, the direct interaction between ANKRD10-2 and MYC could be investigated further to validate their co-activator relationship. Co-immunoprecipitation (Co-IP) experiments were informative, but additional assays, such as chromatin immunoprecipitation (ChIP), could be employed to demonstrate the recruitment of MYC and ANKRD10-2 to target gene promoters, thus reinforcing the proposed regulatory pathway.

Re: Following your suggestion, we conducted ChIP-qPCR experiments (line 260-284, yellow background mark). Based on our previous experimental results showing high MYC protein expression in 5637 cells, we knocked down ANKRD10-1 and ANKRD10-2 separately in 5637 cells, performed enrichment using IgG and MYC antibodies followed by qPCR detection. The results showed that after ANKRD10-2 knockdown, MYC binding to target genes was significantly weakened, while ANKRD10-1 depletion demonstrated minimal effect on MYC's association with target gene promoters (line 537-543, yellow background mark). The specific experimental results are presented in Figure 5i.

8. The Discussion section could benefit from a more explicit acknowledgment of the study's limitations, particularly concerning the potential off-target effects of siRNA and overexpression systems used in vitro. A critical assessment of these limitations would provide a balanced perspective on the findings and suggest directions for future research

Re: We appreciate your valuable feedback regarding study limitations. In response, we have conducted a thorough analysis of potential constraints in our research methodology and incorporated these insights into the discussion section to provide a more comprehensive perspective (line 634-643, yellow background mark).

Reviewer #2 (Comments to the Author):

Yu and colleagues focus on genes differentially expressed between carcinoma and adjacent non-cancerous tissues in bladder cancer (BLCA) to identify the most significant differences in high or low expression between cancer and paracancer tissues

of the TCGA-BLCA cohort.

They identify 5 RNA-binding proteins. Among them, the authors decide to focus on the RNA-binding protein mRNA processing factor (RBPMS), which was downregulated in BLCA patients. Remarkably, elevated RBPMS expression was associated with improved prognosis for BLCA patients.

The authors show that RBPMS impacts on cell migration and invasion and also exerts inhibitory effects on the epithelial mesenchymal transition (EMT) pathway. RNA sequencing experiments revealed that RBPMS depletion in BLCA cells resulted in several alternative spliced transcripts, including the ANKRD10, with increased expression of the ANKRD10-2 isoform.

ANKRD10-2 functioned as a transcriptional co-activator of MYC proteins

The authors conclude that RBPMS suppresses the migration and invasion of BLCA cells by attenuating the MYC pathway through regulation of ANKRD10 splicing.

The study is potentially interesting, but several key details were not provided, including information on the ANKRD10 splicing isoforms. Moreover, the list of the identified splicing events was not provided, and the impact of RBPMS on splicing was not addressed.

Additional points to carefully address are listed below.

Re: Thank you for your thorough review and thoughtful recognition of our work. Your summary effectively captures our study's key findings—from identifying differentially expressed genes in BLCA tissues, to uncovering RBPMS downregulation in BLCA patients and its prognostic implications, to explaining how RBPMS affects cell migration, invasion, and the EMT pathway. We are especially encouraged by your acknowledgment of our discovery that RBPMS inhibits the MYC pathway by regulating ANKRD10 splicing, which influences BLCA cell migration and invasion. We understand your concerns about the need for additional data on ANKRD10 splice isoforms and RBPMS's splicing effects. We will include this essential information in our revised manuscript. Your feedback will help strengthen our study's scientific rigor and completeness. We will address each of your revision suggestions to enhance this work.

1. The authors claim that “Following comprehensive functional inquiries into the cancer upregulated 500 genes in the GO database, they identified five differentially expressed RNA-binding proteins. They authors should report the list of the 500 genes

as a new supplementary table and clarify how these 5 RBPs came up.

Re: Thank you for your suggestion. We have included the gene list in Supplementary Table S6. We input these 500 genes into the GO database (<https://geneontology.org/>) and searched the pathway enrichment results for genes associated with the RNA binding term (GO:0003723). This analysis identified five RNA binding-related genes that we selected for further investigation. We have added a detailed description of this analytical process to the “Results” section (line 328-338, yellow background mark).

2. Bioinformatic analysis upon RBPMS knockdown is not very detailed.

Re: We appreciate your valuable feedback. In response, we conducted KEGG pathway enrichment analysis on the differentially expressed genes identified after RBPMS knockout, with the findings illustrated in Figure S4b. Additionally, we have enhanced the Materials and Methods section with a comprehensive description of our bioinformatics analytical methodology (line 208-223, yellow background mark).

3. Why do the authors decide to focus on ANKRD10?

Re: To identify genes and splicing events regulated by RBPMS, we employed a systematic approach utilizing dual sgRNA-mediated knockdown followed by comprehensive transcriptome and alternative splicing analysis. Our validation studies, which included PCR experiments comparing wild-type and RBPMS knockout cell lines, revealed that ANKRD10 exhibited the most pronounced alternative splicing response to both RBPMS depletion and overexpression. Based on these compelling findings, we selected ANKRD10 as the focus of our subsequent investigations (line 442-444, yellow background mark).

4. The authors claim that RBPMS exerts an inhibitory effect on BLCA metastasis through the cleavage of ANKRD10, how do they reach this conclusion?

Re: Our research demonstrates that RBPMS knockdown leads to enhanced bladder cancer cell migration, as evidenced by comprehensive in vitro and in vivo analyses. Through RNA-Seq analysis and subsequent biochemical validation experiments, we identified ANKRD10 as the primary alternative splicing target regulated by RBPMS. Specifically, we observed a notable increase in ANKRD10-2 splice variant expression following RBPMS depletion. Further investigation revealed that ANKRD10-2 functions as a binding partner of the MYC oncoprotein, facilitating its regulatory

activity on target genes. To establish causality, we performed rescue experiments and found that ANKRD10-2 knockdown partially reversed the enhanced migratory phenotype induced by RBPMS depletion. These experimental findings provide strong evidence that RBPMS regulates BLCA metastasis through its modulatory effect on ANKRD10 splicing.

5. The authors performed RNA-seq analysis to identify transcripts alternatively spliced upon RBPMS knockdown. However, the list of genes and splicing events regulated is not provided. The lists should be provided as supplementary Tables.

Re: Thank you for your suggestion. We have included a comprehensive list of regulated genes and splicing events identified from the RNA-seq data following RBPMS knockout in Supplementary Data 1.

6. How do the authors explain the low overlap between siRBPMS1 and siRBPMS2 in terms of splicing regulation? How does RBPMS1 work in splicing regulation?

Re: We appreciate your thoughtful observation regarding the overlap between sgRBPMS1 and sgRBPMS2. To ensure the robustness of our findings and minimize potential experimental bias, we implemented a dual-knockdown strategy using two independent sgRNA constructs. This methodological approach enabled us to more accurately identify and validate the splicing events under RBPMS regulation. We have thoroughly addressed these experimental considerations in the results and discussion sections (line 426-427, line 634-643, yellow background mark).

7. CLIP experiments should be performed to verify the direct binding of RBPMS to ANKRD10.

Re: In response to your constructive feedback, we performed CLIP-qPCR analysis to examine the molecular interaction between RBPMS and ANKRD10 RNA. Our experimental findings demonstrated a direct binding relationship, with the results documented in Figure 3K (line 286-313, line 454-460, yellow background mark).

8. The scheme of ANKRD10 splicing is not very clear. Moreover, since ANKRD10 splicing has not been described before, a more detailed characterization of these splice variants should be performed. What is the difference in term of protein domain, between the two splicing variants?

It would help the comprehension to indicate the ensembl id.

Re: In response to your feedback, we have enhanced the schematic representation of ANKRD10 splicing patterns and indicated the ensembl id (Figure 3g). Our analysis reveals that both splice variants maintain identical sequences across the initial three exons, specifically in amino acid positions 1-151, which encompasses the conserved ANK repeat domain. Beyond this conserved region, the variants diverge significantly in their exon composition, leading to distinct amino acid sequences from position 152 onward. Notably, ANKRD10-2 exhibits a more compact sequence architecture and consequently a reduced molecular mass compared to ANKRD10-1, as evidenced by its lower migration pattern in Western blot analysis (Figure 3j).

9. Densitometric analysis of the splicing changes should be added to Fig.3F.

Re: In response to this methodological suggestion, we conducted a quantitative analysis of PCR band intensity utilizing ImageJ software. The analysis incorporated β -actin as an internal reference standard for band normalization. To enhance data visualization, we have presented these quantitative findings as a comprehensive heat map in Figure 3f, effectively illustrating the regulatory impact of RBPMS on ANKRD10 splicing patterns (line 442-444, yellow background mark).

10. Two bands are associated with the "inclusion" of ANKRD10 at 216bp (Fig.3f,l); did the authors sequence these bands? Which one did they clone for the IP experiments in Fig.5g)?

Re: We appreciate your thoughtful observation regarding the two bands in the "inclusion" of ANKRD10 at 216bp. We named the longer ANKRD10 transcript variant as ANKRD10-1 (NM_017664.4) and the shorter transcript variant as ANKRD10-2 (NM_001286721.3). It is worth noting that although ANKRD10-1 has longer nucleotide and amino acid sequences, due to the ANKRD10-1 splicing event spanning an exon, the PCR product of ANKRD10-1 is shorter than that of ANKRD10-2. We have added more descriptions and annotations in the manuscript and the splicing diagram of ANKRD10 (Figure 3g) to provide a clearer and more intuitive understanding of the experimental results.

As shown in the following picture, we reanalyzed the original results of RT-PCR analysis. By compared with other genes bands and DNA maker, the higher density bands are more consistent with the "inclusion" bands of ANKRD10 at 216bp (marked

with red arrow), which were used to analyze ANKRD10 splicing, but were not used to clone the plasmids. And the Flag-ANKRD10-1 and Flag-ANKRD10-2 plasmids for the IP experiments were purchased from GeneCopoeia (Guangzhou, China) after sequencing, and we have added this information in the Materials Methods section (line 116-117, yellow background mark).